# Mechanisms of Sperm–Egg Interactions: What Ascidian Fertilization Research Has Taught Us

**DOI:** 10.3390/cells11132096

**Published:** 2022-07-01

**Authors:** Hitoshi Sawada, Takako Saito

**Affiliations:** 1Department of Nutritional Environment, College of Human Life and Environment, Kinjo Gakuin University, Nagoya 463-8521, Japan; 2Graduate School of Science, Nagoya University, Nagoya 464-8602, Japan; 3Department of Applied Life Sciences, Faculty of Agriculture, Shizuoka University, Shizuoka 422-8529, Japan; 4Shizuoka Institute for the Study of Marine Biology and Chemistry, Shizuoka University, Shizuoka 422-8529, Japan

**Keywords:** sperm, fertilization, ubiquitin, proteasome, metalloprotease, lysin, self/nonself recognition, ascidian

## Abstract

Fertilization is an essential process in terrestrial organisms for creating a new organism with genetic diversity. Before gamete fusion, several steps are required to achieve successful fertilization. Animal spermatozoa are first activated and attracted to the eggs by egg-derived chemoattractants. During the sperm passage of the egg’s extracellular matrix or upon the sperm binding to the proteinaceous egg coat, the sperm undergoes an acrosome reaction, an exocytosis of acrosome. In hermaphrodites such as ascidians, the self/nonself recognition process occurs when the sperm binds to the egg coat. The activated or acrosome-reacted spermatozoa penetrate through the proteinaceous egg coat. The extracellular ubiquitin–proteasome system, the astacin-like metalloproteases, and the trypsin-like proteases play key roles in this process in ascidians. In the present review, we summarize our current understanding and perspectives on gamete recognition and egg coat lysins in ascidians and consider the general mechanisms of fertilization in animals and plants.

## 1. Introduction

Sexual reproduction is an important process in living organisms for producing a new individual of the same species with genetic diversity. Sexual reproduction consists of gametogenesis and fertilization, which are accomplished by allogeneic cell–cell interactions and fusion events. Although fertilization is a key process in sexual reproduction, the molecular mechanisms are not yet fully clarified.

Generally, animal fertilization consists of five steps: (1) sperm activation and chemoattraction to the eggs; (2) sperm binding to the proteinaceous egg coat, together with xenogeneic recognition (species-specificity) and allogeneic recognition (self/nonself recognition) processes between gametes; (3) sperm activation and/or acrosome reaction (AR), an exocytosis of the acrosomal vesicle (occasionally, step (3) precedes step (2)); (4) sperm penetration of the egg coat; and (5) gamete fusion [1,2,3,4,5,6,7,8,9] (Figure 1). In the present review, we summarize the historical view and our current understanding of the mechanisms of fertilization. In particular, we will discuss what we have learned from the ascidian fertilization research.

Before discussing the molecular mechanism of ascidian fertilization, we briefly summarize the morphological features and related functions of ascidian egg coats. Ascidians (urochordates), occupying a phylogenetic position between vertebrates and invertebrates, are all hermaphroditic animals. There are solitary ascidians and colonial ascidians, the latter of which proliferate via asexual reproduction [10]. However, even such colonial ascidians utilize sexual reproduction as a reproductive strategy. Solitary ascidians and colonial ascidians are not classified into two independent phylogenetic clades. Most colonial ascidians are ovoviviparous. Therefore, solitary ascidians are more useful for fertilization research than colonial ascidians because of the ease of the fertilization experiments [10]. In the spawning season, solitary ascidians release sperm and eggs nearly simultaneously to the surrounding seawater (SW). However, many, but not all, species, including *Ciona intestinalis* type A (another name for this species is *Ciona robusta* [11], but the name “*Ciona intestinalils*” is used here to avoid confusion.) and *Halocynthia roretzi*, show strict self-sterility. For example, *Ciona savignyi*, *Halocynthia aurantium*, *Styela plicata*, and *Boltenia villosa* are self-sterile, whereas *Phallusia mammillata and Ascidia callosa* are self-fertile [10]. Even in flowering plants, approximately 60% of hermaphroditic angiosperms are self-incompatible (self-sterile) [12]. Therefore, self-sterility in hermaphrodites is certainly beneficial for avoiding inbreeding, but self-fertility may also be advantageous for creating a new organism without other individuals.

*C**. intestinalis* is a cosmopolitan species and is useful for studies of molecular biology because the genome database is readily available and because genome editing experiments are easy to carry out [13,14]. Furthermore, substantial amounts of readily fertilizable sperm and eggs can be easily obtained from gonoducts. *H**. roretzi* is also useful because a large quantity of sperm (approximately 0.1 mL (mean) to 1 mL (max)) and eggs (approximately 1 mL (mean) to 10 mL (max)) can be obtained from each individual by controlling the light conditions and the SW temperature. As this species is aqua-cultured in Onagawa Bay, Japan, for human consumption, it is easy to purchase a large number of individuals. For this reason, we used these two ascidian species for the fertilization research. 

In general, an egg cell is covered by several acellular and cellular coats. The proteinaceous coat is called the vitelline coat (VC) or chorion in ascidians, the vitelline envelope (VE) in sea urchins, and the zona pellucida (ZP) in mammals. The cellular coats surrounding the proteinaceous egg coat are called follicle cells (FCs) in ascidians and cumulus cells in mammals. In sea urchins, the VE is covered with an acellular coat called the jelly layer, which induces AR [2].

An ascidian egg has unique cells called test cells (TCs) in the perivitelline space that are involved in the formation of larval tunics during development [10] (Figure 1). Although TCs are not directly involved in fertilization, TCs, together with egg cells, appear to transcribe two mRNAs encoding the C-terminal region of vitellogenin during oogenesis [15]. After proteolytic processing, two C-terminal fragments of the vitellogenin attach to the VC, which in turn participate in gamete interaction via the interaction with sperm trypsin-like proteases (HrProacosin and HrSpermosin) in *H*. *roretzi* [15,16,17,18] (for details, see Section 6.2).

FCs have many vacuoles and are responsible for egg flotation in SW [19]. There is a report suggesting that FCs release sperm chemoattractants [20]. Furthermore, FCs are thought to be involved in polyspermy block by releasing glycosidases that hydrolyze or bind to the sugar moiety of the VC glycoproteins functioning as a sperm receptor [21]. Interestingly, FCs are essential for fertilization in *Halocynthia roretzi* [22] but not in *Ciona intestinalis* [10]. When most, but not all, of the FCs are removed by pipetting through a small-bore glass pipette, the spermatozoa can only penetrate through the FC-attached VC region in *H*. *roretzi* [22]. FC-free eggs, or even the eggs suspended in FCs, cannot be fertilized [22]. These results indicate that the attachment of the FCs to the VC is indispensable for sperm penetration of the VC, although the reason is still unknown. In contrast, TCs are not needed for fertilization, as the VC-removed eggs, having no FC, VC, or TC, can fuse with self and nonself sperm. In addition, fertilization experiments using “mosaic” eggs, whose original FCs are replaced with those of a different individual, clarified that the VC plays a key role in self/nonself recognition in *H*. *roretzi* [22]. Although FCs are not directly involved in self/nonself recognition during fertilization, FCs are responsible for the acquisition of self-sterility during oocyte maturation in both *H*. *roretzi* and *C*. *intestinalis* (for details, see Section 4).

Ascidian spermatozoa contain a single cigar-shaped mitochondrion attached to the side of the sperm head. Upon sperm binding to the VC, the sperm undergo a unique morphological change called the “sperm reaction” [23,24] (Figure 1). In this reaction, a sperm mitochondrion swells and becomes spherical, followed by sliding toward the tip of the tail and eventual shedding (Figure 1). Ascidian sperm have a tiny acrosome at the tip of the sperm head. However, the timing of AR is still debated [25,26] (see Section 5).

## 2. Ascidian Sperm Activation and Chemotaxis

Sea urchin spermatozoa are activated by the sperm activation peptide (SAP) contained in the jelly layer [1]. It is reported that at least one of the SAPs, called “resact”, consisting of 14 amino acid residues, has a chemoattracting activity for *Arbacia punctulate* spermatozoa [27]. It is reported that resact triggers a rapid and transient increase in intracellular cGMP, followed by transient Ca^2+^ influx [27,28]. However, the detailed mechanisms of sperm chemotaxis are poorly understood.

In contrast to sea urchin spermatozoa, ascidian spermatozoa are immotile when dispersed in SW [29,30]. However, the sperm undergo vigorous movement and are attracted toward the egg by SAAF (sperm-activating and -attracting factor), which is released from the egg cell but not the FCs, as suggested previously [18]. SAAF is known to be species-specific or genus-specific [29,30]. The structure of the SAAF of *C**. intestinalis* (type A), called Ci-SAAF, is a sulfated steroid 3,4,7,26-tetrahydroxycholestane-3,26-disulfate [31], and the SAAF of *Ciona savignyi* is identical to Ci-SAAF [29,30,31]. The structures of the SAAF of another ascidian species have also been reported [29,30]. The mode of action of Ci-SAAF in chemotaxis has been well studied on a molecular basis. A transient intracellular Ca^2+^ increase is induced by the SAAF gradient. It has been reported that “Ca^2+^ bursts induced around a local minimal SAAF concentration during circulation trigger a sequence of flagellar responses comprising quick turning followed by straight swimming to direct spermatozoa efficiently toward eggs” [32] (see Figure 2A). Transiently increased Ca^2+^ binds to sperm calaxin, a Ca^2+^ sensor, which suppresses the microtube sliding of the outer-arm dynein, resulting in the generation and propagation of Ca^2+^-induced asymmetric flagellar bending [33]. Recently, SAAF-binding protein was identified as sperm plasma membrane calcium/calmodulin-dependent calcium-ATPase (PMCA), which is involved in Ca^2+^ efflux [34]. PMCA activity is stimulated by SAAF, and a PMCA inhibitor inhibits the chemotactic behavior of sperm, whereas store-operated calcium channels are involved in Ca^2+^ influx, which is indicated by specific inhibitors [35]. It has been proposed that when the SAAF gradient is descended, the Ca^2+^ efflux ability of the PMCA is decreased by the dissociation of the SAAF, resulting in a transient increase in the intracellular Ca^2+^ concentration, which is mediated by the store-operated calcium channels [29,34,35]. Although the structures of SAAF from several species have been determined [29,30], the occurrence and structure of *H*. *roretzi* SAAF have not been reported.

## 3. Sperm Binding to the VC in Ascidians

After sperm chemoattraction to the eggs, the spermatozoon binds to the egg coat. The mechanisms of species-specific sperm binding to the VE are well known in sea urchins and abalones. In sea urchins, the species-specific binding of sperm to the VE is mediated by the sperm acrosomal 30-kDa protein called “bindin”, which has a variety of amino acid sequences among species [2,36], and its binding partner EBR1 (egg bindin receptor 1) [37] on the VE. EBR1 is a large (~350-kDa) ADAMTS-family protein consisting of a repeated structure of CUB domains and thrombospondin-type-1 repeats [2,37].

On the other hand, abalone nonenzymatic lysin can bind to the receptor on the VE called VERL (vitelline envelope receptor for lysin) [2]. Abalone lysin functions not only as a species-specific binding protein to the VE but also as a lysin to make a small hole for the sperm passage of the VE by swelling the filamentous network of VERL. Red abalone VERL, comprising 3722 amino acids, consists of 22 tandem repeats of 153 amino acids. Each repeat shows structural homology to the N-terminal ZP domain 1 (ZP-N1) of mouse ZP2. Repeats 1–2 are subjected to positive selection, while repeats 3–22 are homogenized by concerted evolution [2]. The interaction between abalone lysin and VERL is well studied on 3D structural bases. Abalone VERL forms a rod-like dimer mainly supported by hydrogen bonds, and repeat 2 of VERL is responsible for species-specific binding between lysin and VERL [2,38]. Lysin species-specifically binds to repeat 2 of VERL by hydrophobic interactions, resulting in the exposure of highly basic residues of lysin to the surface. Then, lysin sequentially binds to repeat 3 and subsequent repeats, causing an electrostatic repulsion of the VERL-lysin complex, which creates a hole for sperm passage. In contrast to these findings, the molecular mechanisms of species-specific gamete binding are not well known in ascidians.

### 3.1. Gamete Binding in Ciona intestinalis

In *C*. *intestinalis*, we performed a proteomic analysis of VC proteins by LC/MS/MS [39]. The most abundant protein in the VC is an apolipoprotein B-like (ApoB-like or vitellogenin-like) protein [39]. Its sperm-side binding partner was identified as a 70-kDa sperm protein by Far Western blot analysis. The VC contains many ZP proteins [39]. Among eleven ZP proteins, CiVC57 is the most abundant, comprising a signal peptide, a von Willebrand factor domain, 24 EGF-like repeats, a ZP domain, and a C-terminal transmembrane domain [39]. On the other hand, a cysteine-rich secretory protein (CRISP), designated CiUrabin, is one of the major sperm surface proteins [40]. The 40-kDa CiUrabin consists of a pathogenesis-related (PR) domain and a glycosylphosphatidylinositol (GPI)-anchor attachment site [41]. The *CiUrabin* gene is specifically expressed in the testis, and the CiUrabin protein is localized on the surface of the sperm head, as revealed by immunocytochemistry and proteomic analysis [40,41]. As the sperm CRISP family proteins are involved in mammalian fertilization [42] and as HrUrabin [43], an *H*. *roretzi* homolog of CiUrabin, can bind to HrVC70 [44,45], an *H*. *roretzi* homolog of CiVC57, whether CiUrabin can interact with CiVC57 was investigated using Far Western blot analysis. As expected, CiVC57 was found to bind to CiUrabin [41]. Therefore, primary binding between sperm and the VC must be mediated by the interaction between CiUrabin and CiVC57 [6,41] (Table 1).

In addition to the protein–protein interaction, sperm glycosidase and the carbohydrate moiety of the VC also play a key role in gamete interaction. It is notable that the sperm α-L-fucosidase participates in sperm binding to the L-fucosyl residues of the VC glycoproteins [46,47,48]. The synthetic substrates (p-nitrophenyl or 4-methylumbellyferyl α-L-fucoside) and their β-anomers functioning as competitive inhibitors inhibited sperm binding to the VC of glycerol-treated eggs. α-L-fucosidase was purified from *C*. *intestinalis* sperm [47]. The molecular mass was estimated to be 105 kDa by SDS–PAGE and 112 kDa by HPLC. It has been reported that five monoclonal antibodies against purified α-L-fucosidase inhibit sperm binding to the VC. Immunostaining using two monoclonal antibodies revealed localization at the sperm head, mostly at the tip and probably at the surface. While *C*. *intestinalis* sperm α-L-fucosidase had an optimum pH of ~3.9, it showed 2% or less of maximum hydrolytic activity in normal SW [48]. In addition, spermatozoa bound to the VC in normal SW detach under acidic conditions (~optimum pH) within 7 min at 20 °C but not at 0 °C [47,48]. These results suggest that α-L-fucosidase at the tip of the sperm head is involved in the primary binding of sperm to the fucosyl glycoprotein(s) on the VC in *C**. intestinalis* [48]. The 140-kDa, 125-kDa, and 78-kDa fucosyl glycoproteins on the VC are candidate sperm receptors [49]. The fucosyl glycoproteins appear to be localized on tufts of the outer surface but not the inner surface of the VC, as revealed by electron microscopy using fucose-binding protein and fucose-ferritin [50]. These VC glycoproteins are potential receptors for sperm α-L-fucosidase (Table 1).

### 3.2. Gamete Binding in Halocynthia roretzi

The 70-kDa major VC protein HrVC70 functions as a sperm receptor in *H. roretzi* [44,45]. HrVC70 is localized on the outer surface of the VC, exposed between FCs, where sperm can specifically bind [5,51] (Figure 2B). HrVC120, a precursor of HrVC70 comprising 12 EGF-like repeats, consists of a signal peptide, 12 EGF-like repeats, one incomplete EGF-like domain, and a C-terminal ZP domain [5,44]. HrVC120 is expressed in the gonad but not in the other tissues or cells tested [44]. After the processing of the signal peptide, HrVC120 is processed at the C-terminal side of Arg^668^, which is catalyzed by an FC trypsin-like protease, which is most likely HrOvochymase [52]. HrOvochymase is a polyprotease consisting of a signal peptide, a trypsin-like protease domain, two CUB domains, a trypsin-like protease domain, three CUB domains, and a serine protease domain missing a typical catalytic triad [52]. This protease, mainly localized in the FC, is responsible for the expansion of the perivitelline space and for the acquisition of self-sterility by the processing of HrVC120 into HrVC70 during oogenesis [52,53]. The HrOvochymase and HrProacrosin genes appear to be identical, but the mRNA species look different [18,52].

The Far Western blot analysis revealed that sperm HrUrabin is a binding partner of HrVC70 [43]. This protein is classified into the CRISP family [42]; it contains a PR1 domain and a GPI-anchor-attachment site and was designated HrUrabin [43]. HrUrabin is a GPI-anchored sperm surface protein, as the treatment of intact sperm with PI-PLC reduced the molecular size of HrUrabin. An anti-HrUrabin antibody revealed the existence of a closely related 50-kDa isoform (HrUrabin-L) by Western blot analysis, and its amino acid sequence is very similar to the sequence of HrUrabin, albeit with C-terminal 6 repeats of hexapeptide E(A/V/G)(A/V)DGD in HrUrabin-L [43]. Although the role of the C-terminal repeat is not known, it is interesting to note that HrUrabin-L cannot interact with HrVC70. Most likely, the C-terminal repeat may interfere with the interaction between HrVC70 and HrUrabin. Intriguingly, deglycosylated HrUrabin with PNGase-F cannot bind to HrVC70. Therefore, the N-linked sugar moiety must be necessary for the binding of HrUrabin to HrVC70. Although X-ray crystallographic analyses of HrVC70 and HrUrabin have not been performed, the 3D structure of HrVC70 was predicted by AlphaFold2 [54] (Figure 3). Immunocytochemistry showed the patch-like localization of HrUrabin throughout the sperm [43]. The anti-HrUrabin antibody strongly inhibited the binding of self and nonself sperm to HrVC70-agarose beads and potently inhibited fertilization. However, there was no apparent difference between the binding ability of HrVC70 to HrUrabin self sperm and that to HrUrabin nonself sperm, according to Far Western blot analysis. These results indicate that the interaction of glycosylated HrUrabin and HrVC70 is evident, but this interaction may not be directly involved in self/nonself recognition.

Yeast two-hybrid screening revealed that two ZP proteins on the VC can bind to HrVC70 [53]: a 90-kDa HrVLP-1 (VC70-like protein) and a 54.8-kDa HrVLP-2, both of which contain a signal peptide and 5 and 1 EGF-like repeat(s), respectively, and a ZP domain. HrVLP-1 and -2 may interact with HrVC70, forming a fibrous network, probably via the ZP domains, similar to mammalian ZP1–3 [8]. In addition to VC proteins, a type-II transmembrane protein called HrTTSP-1 was identified as a sperm-side binding partner for HrVC70. This 337-kDa protein consists of an N-terminal-side transmembrane domain, 23 EGF-like domains, 2 ricin-B-lectin domains, and a C-terminal-side serine protease domain [53]. The protein–protein interaction between HrVC70 and HrTTSP-1 was confirmed by a pulldown assay. Taken together, sperm HrTTSP-1 and HrUrabin are thought to be the main binding partners of HrVC70.

Similar to *C. intestinalis*, sperm 52.4-kDa *α*-L-fucosidase and the fucosyl glycoproteins in the VC are thought to be involved in the interaction between sperm and the VC, as anti-*α*-L-fucosidase antiserum inhibited fertilization [55]. Although the target of sperm *α*-L-fucosidase has not been identified, fucosylated HrVC70 may be a candidate binding protein for sperm *α*-L-fucosidase. There is a consensus sequence (C_2_XXGG(S/T)C_3_) for O-fucosylation in the EGF-like domain [56], where C_2_ and C_3_ indicate the second and third Cys residues, respectively, in the EGF-like domain. O-fucosylation includes monosaccharide modification, tetra-saccharide modification (NeuAc*α*2→6Gal*β*1→4GlcNAc*β*1→3Fuc*α*1→O-Ser), or its 2- or 3-residue intermediate [56]. There are five potential O-fucosylation sites in the EGF domains of HrVC70 (Figure 3A). LC/MS/MS was used to investigate whether these potential sites were glycosylated. Among the five potential O-fucosylation sites, at least Thr^362^ in EGF-7 was fully O-fucosylated: there was no free Thr^362^ or disaccharide- or trisaccharide-modified Thr^362^. Partial modification by fucose but not by disaccharides or trisaccharides was detected at Ser^143^ in EGF-3. Partial modification of fucose and trisaccharides (Fuc, HexNAc, and Hex) but not disaccharides was observed in Thr^254^ in EGF-5. Partial modification by trisaccharides (Fuc, HexNac, and Hex) but not by monosaccharide or disaccharide chains was observed at Ser^470^ in EGF-9. In contrast, no fucosylation or disaccharide or trisaccharide modifications were detected at Ser^578^ in EGF-11 (Figure 3). Fucose-specific lectin from *Aspergillus oryzae* specifically reacted to HrVC70, as shown by SDS–PAGE, followed by lectin blotting. This lectin strongly inhibited the fertilization of *H. roretzi* in a concentration-dependent manner (Sawada, Sawa, Itoh, and Kawasaki, unpublished data). These results indicate that HrVC70 is a candidate binding protein for sperm *α*-L-fucosidase as this protein contains a nonreducing terminal fucose residue at least at Ser^143^ and Thr^362^. As discussed later, it is speculated that sperm *α*-L-fucodiase will not bind to HrVC70 when N-acetylglucosaminidase (GlcNAcase), which is released from eggs upon fertilization, binds to the trisaccharide-chain-modified region of HrVC70 (Figure 3A). This may be related to the polyspermy block in *H. roretzi*.

## 4. Self/Nonself Recognition between Sperm and Eggs in Ascidians

Many ascidian species show strict self-sterility or preference for nonself fertilization [6,10,57,58,59,60,61,62]. Upon sperm binding to the VC, the sperm undergo a self/nonself recognition process [6,12]. Regardless of the lack of adaptive immunity in ascidians (Urochordate), how can ascidian sperm distinguish between self and nonself eggs? This has been a long-standing enigma for more than a hundred years [58].

### 4.1. Self/Nonself Recognition in Ciona intestinalis

Ciona intestinalis has been widely used for self/nonself recognition studies during fertilization, as only nonself sperm can tightly bind to the VC of glycerol-treated eggs, which are reported to allow sperm binding to, but not passage of, the VC [62]. Several groups have explored factors responsible for self/nonself recognition in gamete interactions in this species [63,64,65,66,67,68]. One of the interesting common features is that FCs are responsible for the acquisition of self-sterility during the late stage of oogenesis as FC-free immature oocytes cannot acquire self-sterility during oogenesis [69,70,71,72]. The proteasome inhibitor clasto-lactacystin *β*-lactone inhibited the acquisition of self-sterility during GVBD but not after GVBD, suggesting that the FC proteasome functions before GVBD [71]. In addition, two anti-hsp70 antibodies (a commercially available antibody and the antibody raised against peptides corresponding to the C-terminal sequence of the isolated cDNA of Ci-hsp70) inhibited the acquisition of self-sterility during oogenesis. Immunocytochemistry revealed the localization of hsp70 on the surface of FC. As the (immuno)proteasome is involved in the trimming of antigenic peptides, which are embedded into MHC for antigen presentation, and as the hsp70 gene, belonging to the MHC class-III gene, is thought to be an ancestral gene of MHC class-I and -II [72], autologous peptides produced by the FC proteasome may be associated with hsp70 and exposed to the surface of FCs during oogenesis by analogy to antigen presentation by MHC. These peptides might be transferred to a putative receptor protein on the VC, which in turn plays a role in self/nonself recognition between gametes [71,72]. However, the putative autologous peptides associated with Ci-hsp70 and the sperm recognition proteins of these putative peptides have not been identified.

Kawamura and his collaborators attempted to identify a putative allorecognition factor from a different angle [63,64]. They found an activity in the supernatant of acid-treated eggs that inhibits nonself sperm from binding to the VC of glycerinated eggs [64]. They succeeded in identifying two factors: one is a non-allorecognizing glucose-enriched inhibitor against sperm binding to the VC, and the other is a Glu/Gln-enriched peptide modulator; they play roles as cofactors in the allorecognition of sperm receptors [23,64]. However, the molecular mechanism remains elusive.

Using *C. intestinalis* type B, Khalturin and his collaborators identified several genes expressed in oocytes and FCs that are highly polymorphic among individuals. They revealed the occurrence of highly polymorphic ZP proteins and sushi domain-containing protein vCRL1. They proposed that these proteins might be involved in fertilization and allorecognition in *C. intestinalis* type B [65,66]. However, their genetic analysis of and gene-knockdown experiments on vCRL1 showed that vCRL is not involved in self/nonself recognition during fertilization but is likely to be involved in the establishment and maintenance of the blood system [73,74]. Importantly, their genetic analysis also supported that s/v-Themis gene pairs (described below), but not the vCRL1 gene, are responsible for self-sterility in *C. intestinalis* type B [74].

In contrast to the above biochemical approaches, genetic analyses were also conducted by several groups, including our laboratory. Self-sterility in *C*. *intestinalis* was first genetically studied by Thomas Hunt Morgan more than a hundred years ago [58,59,60,61,62]. Murabe and Hoshi [75] and our laboratory [67] repeated Morgan’s experiments to examine whether self-fertilized F1 siblings show cross-sterile combinations [67,75]. They confirmed Morgan’s observation, i.e., that there is a one-way nonself-sterile combination between self-fertilized F1 siblings, which is scarcely observed under natural crossing. What is one-way nonself-sterility? Sperm of individual A can fertilize the eggs of individual B, but not vice versa. From these results, Morgan concluded that the self-sterility system is genetically controlled. He hypothesized that the “male” (sperm-donor) self-sterility-responsible gene(s) are expressed in haploids, while the “female” (egg-donor) self-sterility-responsible gene(s) are expressed in diploids. This is called the “haploid sperm hypothesis”. In this case, heterozygous individuals (A/a) release both A-sperm and a-sperm, and both sperm mixtures can fertilize homozygous a/a- and A/A-eggs because A-sperm can fertilize a/a-eggs and a-sperm can fertilize A/A-eggs. On the other hand, homozygous individuals (A/A or a/a) release A-sperm or a-sperm alone, respectively, which cannot fertilize the eggs of heterozygous individuals (A/a-eggs) because both A- and a-sperm receptors exist in the VC and are recognized as self eggs. Based on these criteria, our laboratory carried out crossing experiments between self-fertilized F1 siblings and searched for a one-way cross-sterile combination [67]. Then, “male” and “female” individuals showing one-way nonself-sterility were subjected to genomic analyses at 70 gene markers in 14 chromosomes by PCR to determine whether they were homozygous or heterozygous [67]. As a result, two loci (locus A on chromosome 2q and locus B on chromosome 7q) were identified as self-sterility-responsible loci. Among approximately 20 genes in locus A, only one gene product (fibrinogen-like protein) was detected in the VC by proteome analysis [41], and a polymorphic gene expressed in the testis was identified as a candidate sperm-side self-sterility-responsible factor [67]. These gene pairs were designated *s*(*sperm*)*-Themis-A* and *v*(*vitelline coat*)*-Themis-A*, and two similar gene pairs were found in locus B and named *s***/***v-Themis-B* and *s***/***v-Themis-B2* [67,68]. By detailed genetic analysis and close inspection, we demonstrated that three multiallelic gene pairs must play a pivotal role in self-sterility: egg-side genes (*v-Themis-A*, *v-Themis-B*, and *v-Themis-B2*) and sperm-side genes (*s-Themis-A*, *s-Themis-B*, and *s-Themis-B2*) [68]. Interestingly, *v-Themis* genes are encoded in the first intron of *s-Themis* genes in the opposite direction [67,68]. The sperm-side self-sterility-responsible genes *s-Themis-A*, *-B*, and *-B2* showed homology to mammalian *PKD1* or *PKDREJ*, both of which contain a hypervariable region (HVR), a receptor for egg jelly (REJ), a G protein-coupled receptor proteolysis site (GPS), lipoxygenase homology 2 (LH2) domain, and 5-pass (in the case of *s-Themis-A*) or 11-pass (in the case of *s-Themis-B* and *s-Themis-B2*) transmembrane (TM) domain. Notably, *s-Themis-B* and *s-Themis-B2* possess a cation channel (polycystic kidney disease (PKD) channel) domain in their C-terminal regions [67,68]. When three multiallelic gene pairs were matched (i.e., the same haplotypes), the eggs could not be fertilized [67]. Furthermore, when *s***/***v-Themis-A* genes and *s***/***v-Themis-B***/***B2* genes were impaired by genome editing using TALEN, the *s***/***v-Themis-B***/***B2*-targeted individual sperm could fertilize the self eggs. *s-Themis-A*-targeted sperm also fertilized the self eggs, but a significant delay was observed in self-fertilization compared with *s-Themis-B***/***B2*-targeted sperm [68]. These data indicate that *s-Themis-A*, *B*, and *B2* are essential for exhibiting self-sterility [68]. Furthermore, *s-Themis-B***/***B2* appears to be more directly involved in allorecognition than *s-Themis-A*. Further studies are necessary to clarify the mechanisms of the allospecific protein–protein interactions and 3D structures of the complexes between s-Themis and v-Themis.

After the sperm recognize the VC as self, a drastic and acute Ca^2+^ influx called the self-incompatibility response occurs in the sperm [76]. This dramatic self-incompatibility response causes vigorous sperm movement and detachment from the VC or eventual sperm quiescence. Such a response is prevented by lowering the external Ca^2+^ concentration in SW, suggesting that Ca^2+^ influx may transpire after self recognition. Notably, self-sterility is cancelled under the conditions of low Ca^2+^ (~1 mM) concentration, suggesting that the inhibition of the increase in intracellular Ca^2+^ concentration is sufficient to cancel self-sterility [77]. In other words, the increase in intracellular Ca^2+^ concentration may trigger the self-sterility response within sperm. Notably, this system is very similar to the self-incompatibility system in Papaveraceae [12] (Table 2). In Papaveraceae, the pollen PrpS recognizes the stigmatic PrsS as self, and Ca^2+^ influx occurs in the pollen, resulting in programmed cell death. In this system, the gene pair of *PrpS* and *PrsS* is highly polymorphic and contiguously localized. Therefore, the molecular mechanism of the self-sterility system (s/v-Themis-A, B, and B2) in *C*. *intestinalis* is very similar to that of the self-incompatibility system in flowering plants, particularly the self-incompatibility system in Papaveraceae. By analogy, the increase in sperm intracellular Ca^2+^ concentration induced by attachment to the self-VC may trigger sperm cell death because the increased Ca^2+^ concentration is too high to maintain sperm viability.

In mice, there is a homolog of s-Themis called PKDREJ, a candidate sperm-side receptor [7,78]. Although *PKDREJ*-KO mouse sperm are fertile, a certain delay (more than 2 h) was observed in the arrival time to the egg/cumulus complex after mating, and a certain delay in ZP-evoked AR was observed compared with normal sperm [78]. Therefore, PKDREJ may control the timing of fertilization via its effects on sperm transport and ZP-evoked AR [78]. Interestingly, PKDREJ is reported to evolve under positive selection and shows polymorphisms among individuals in humans and primates [79]. These features led us to speculate that mammalian sperm PKDREJ, a homolog of s/v-Themis, may be related to fertilization efficiency between gametes with genetic proximity.

As described previously, the acid extract of *C*. *intestinalis* eggs shows the ability to inhibit nonself sperm from binding to the VC [64]. To explore a factor involved in this process, the acid extract of the VC was subjected to LC/MS/MS analysis. Whereas v-Themis proteins were not efficiently extracted by acidic conditions, a “v-Themis-like” protein was found to be efficiently solubilized by this condition. This protein has a fibrinogen-like domain, similar to v-Themis-A, -B and -B2, but few or no apparent polymorphisms were observed among individuals [80]. Although this protein may not be involved in self-sterility, it may be involved in the assembly of v-Themis-A, -B, and -B2 on the VC and/or in gamete binding by interacting with sperm trypsin-like proteases [80].

### 4.2. Self/Nonself Recognition in Halocynthia roretzi

Self-sterility in *H. roretzi* is very strict. In fact, when two individuals’ eggs were differentially stained by neutral red or Nile blue, mixed together, and inseminated, only nonself sperm could fertilize the eggs (unpublished). This suggests that even spermatozoa potentially interacting with the VC of nonself eggs cannot fertilize the next eggs of the same individual. In contrast, when the eggs were treated with acidic SW (pH ~ 2.5) for a short period (~1 min), they became self-fertile [81]. Immature oocytes and VC-free eggs are also self-fertile [45,81]. From these results, we speculated that a putative allorecognition factor may be attached to the VC during oocyte maturation and that a putative factor may be detached from the VC or irreversibly denatured by acidic conditions. To test this possibility, VCs from immature oocytes and mature eggs were subjected to SDS–PAGE to compare the VC components. The results showed that the amount of HrVC70 in the VC of mature eggs was considerably higher than the amount in the VC of immature oocytes [45]. Furthermore, HrVC70, but not the other VC components, was efficiently or almost specifically solubilized from the isolated VC by 1 or 5 mM HCl, but not by SW (pH ~ 8.2), suggesting that HrVC70 is insoluble under natural SW but is soluble in acidic conditions. When isolated HrVC70 was immobilized to agarose beads and incubated with self and nonself sperm, the number of nonself sperm bound to HrVC70-agarose was significantly higher than the number of self sperm bound to HrVC70-agarose. Furthermore, HrVC70 isolated from nonself eggs more efficiently inhibited fertilization than HrVC70 isolated from self eggs [45]. From these results, and because HrVC70 shows high polymorphisms among individuals at restricted sites between the third and fourth Cys residues and between the EGF-domain-connecting regions, while even a single amino-amino acid substitution in EGF-like repeat regions in Notch protein causes Notch-signaling diseases [82], HrVC70 may be a promising candidate for allorecognition in the fertilization of *H. roretzi*. Although it is still unclear whether the amino acid substitution in HrVC70 is truly responsible for allorecognition during gamete interaction in *H. roretzi*, all of the biochemical data obtained thus far support our working hypothesis that HrVC70 is a promising candidate responsible for self/nonself recognition in *H. roretzi* (Figure 2B and Figure 3).

In *H. roretzi*, trypsin-like protease in FCs plays a key role in the acquisition of self-sterility during oogenesis for the following reasons. Immature oocytes in the gonads are self-fertile, but they become self-sterile after germinal vesicle breakdown (GVBD) [81]. This process is inhibited by microbial and proteinaceous trypsin inhibitors [83,84]. Furthermore, the acquisition of self-sterility in FC-removed immature oocytes is mimicked by adding trypsin. These results indicate that a trypsin-like protease, which is released from FCs, may play an extracellular key role in the activation of self-sterility-responsible factors on the VC. As discussed later, this putative precursor protein appears to be HrVC120, which is processed to HrVC70 via proteolysis at Arg^668^ by FC trypsin-like protease, most likely by FC HrOvochymase [52,85].

In the self-incompatibility system in Solanaceae and Rosaseae, pistil S-RNase is thought to penetrate into an elongating pollen tube, where only nonself S-RNase is ubiquitinated by SLF/SFB (S-locus F-box) and degraded by the 26S proteasome in the pollen tube, resulting in hydrolysis of the RNA by the surviving self S-RNase (Table 2) [12]. As discussed in Section 6, nonself HrVC70 is thought to be ubiquitinated by a ubiquitinating enzyme complex [86], which is secreted from sperm upon sperm reaction. Ubiquitinated HrVC70 must be degraded by the sperm 26S proteasome [44,87,88]. The *H. roretzi* self-sterility system seems very similar to the self-incompatibility system in Solanacear and Rosaceceae [12]. This led us to speculate about the occurrence of a common reproductive mechanism shared by animals and plants [12].

Halocynthia aurantium is in the same genus of ascidian as *H. roretzi*, and it inhabits the shallow seas around northern Japan (Hokkaido) to North America. This species has an HrVC70 homolog called HaVC80, an 80-kDa protein consisting of 13 EGF-like repeats, whose basic structure is very similar to HrVC70 except for having two 8th EGP domains [89]. Most likely, the 8th domain of HrVC70 may have been duplicated during evolution. The exon/intron organization is also very similar, and the variable regions are also similar but add a region between the 1st and 2nd Cys residues. Analogous to HrVC70, HaVC80 may be involved in sperm–egg interactions. However, the binding partner for HaVC80 has not been identified, whereas a homologous gene to HrUrabin is present in the genome database of H. aurantium.

As three multiallelic gene pairs of s-Themis and v-Themis play a key role in self/nonself recognition in *C. intestinalis*, we explored the homologous gene pairs in *H. roretzi*. In the genome database ANISEED [14], at least four alleles of s/v-Themis gene pairs were found in the *H. roretzi* genome database. However, as it takes 3 years for *H. roretzi* to grow to an adult, carrying out genetic analysis in natural sea conditions is very difficult. However, these gene pairs exhibited a substantial polymorphism, albeit to a lesser extent compared with *C. intestinalis* s/v-Themis. These results imply that s/v-Themis gene pairs may also be involved in self/nonself recognition in *H. roretzi*.

## 5. Acrosome Reaction (AR) and Sperm Reaction

Sperm AR is an exocytosis of an acrosomal vesicle elicited by egg investments, which is widely observed in many animals, including sea urchins and mammals. In sea urchins or many other marine invertebrates, sperm protrude an “acrosomal process” at the tip of the sperm head during AR, which is produced by actin polymerization beneath the sperm inner acrosomal membrane [1]. Fucose sulfate polymer (FSP) in egg jelly interacts with REJ (receptor for egg jelly), particularly suREJ1, on the sperm head surface, which elicits the AR [90]. In addition to FSP, a sialoglycoprotein and speract (SAP) appear to be involved in this process at lower pH (~pH 7) [91]. It is interesting to note that the structural difference in the carbohydrate moiety of FSP is responsible for species specificity in three species of sea urchins [92]. It is also notable that egg-jelly-induced AR, but not Ca^2+^-ionophore-induced AR, is inhibited by a proteasome inhibitor, suggesting that the proteasome may be involved in AR in sea urchins [93].

In the starfish *Asterias amurensis*, Hoshi and Matsumoto and their colleagues have reported that high molecular weight glycoproteins, called ARIS (acrosome rection-inducing substance), Co-ARIS (cofactor of ARIS, a sulfated steroid saponin), and asterosap (oligopeptide, SAP) are involved in the induction of AR [94]. It is proposed that asterosap first interacts with a receptor, guanylate cyclase, in the sperm tail, which stimulates the synthesis of cGMP, [Ca^2+^]i and [H^+^]i. cGMP induces the increase in [Ca^2+^]i, which appears to be mediated by a K^+^-dependent Na^+^/Ca^2+^ exchanger. ARIS alone is sufficient to induce AR in high [Ca^2+^] SW or high pH SW (pH ~ 9.5) but not in regular SW, in which Co-ARIS or asterosap is required. It is also known that ARIS and asterosap can induce the sustained increase in [Ca^2+^]i and the elevation of [cAMP]i [94].

In mammals too, sperm undergo the AR during the passage of the cumulus layer and upon binding to the zona pellucida [1]. Although it has long been believed that the AR takes place upon sperm binding to the zona pellucida, live-imaging experiments have revealed that most fertilizing mouse spermatozoa undergo the AR before contact with the zona pellucida, probably during the passage of the cumulus layer [95].

In contrast to sea urchins or mammals, ascidian sperm do not have a typical acrosome, but a very small acrosome at the tip of the sperm head. Therefore, it is not easy to assay AR. The timing of AR during fertilization is still debated. It remains elusive whether AR occurs upon sperm binding to VC [25,26] or after penetration prior to gamete fusion [26]. It cannot be ruled out that sperm can penetrate the VC without undergoing AR through the hole, which had been made by previously penetrated sperm, whereas most sperm may undergo AR on the VC. Electron microscopic observation will not answer this question. Live imaging experiments using GFP-tagged acrosomal protein-expressing sperm are necessary to clarify the timing of AR in ascidians. Instead of AR, the ascidian sperm undergoes “sperm reaction”, which is characterized as vigorous movement on the VC, mitochondrial swelling and sliding to the tip of the tail, and eventual shedding. This requires Ca^2+^ but not Na^+^ and is mimicked by alkaline SW (pH 9.5) or the Ca^2+^ ionophore A23187. This mitochondrial translocation is thought to be mediated by actin-myosin-driven movement, as cytochalasin B inhibits this reaction and anti-actin and anti-myosin antibodies show the existence of these proteins on the tail and mitochondrial regions by immunocytochemistry [96,97]. Sperm reaction appears to correspond to sperm activation, but it is not clear whether or not “sperm-reacted” sperm undergo AR.

## 6. Sperm Egg-Coat Lysins

### 6.1. Historical View of Lysins

Sperm VC lysin research dates back to 1930 [98,99,100]. The “lysin” activity was first suggested by Yamane in rabbit spermatozoa [98]. He reported that rabbit sperm extract can affect the zona pellucida and disperse the cumulus cells [99,100]. His findings implied that sperm glycosidase (hyaluronidase) and protease (acrosin) may be lysins in mammalian spermatozoa. Since then, it has long been believed that the acrosomal trypsin-like protease acrosin [5,6,12,23] is a lytic agent lysin that is released by acrosomal exocytosis and digests the ZP, creating a small hole for the sperm passage of the ZP [101,102]. However, such a function of sperm acrosin has been debated, as sperm binding to the ZP but not sperm penetration of the ZP was strongly inhibited by several trypsin inhibitors [103]. Furthermore, it has been reported that purified boar acrosin can degrade [^125^I]-labeled heat-solubilized ZP proteins (90-kDa, 65-kDa, and 55-kDa proteins) and that boar acrosin can bind to [^125^I]-heat-solubilized ZP proteins, which are inhibited by DFP, an irreversible serine protease inhibitor [104]. In 1994, Baba and his colleagues showed that mouse acrosin is not essential for fertilization or sperm penetration of the ZP by using *acrosin-*knockout (KO) mice [105]. However, an approximately 30 min delay was observed in the sperm penetration process of the ZP in *acrosin*-KO sperm [105]. Therefore, it is currently thought that acrosin is not essential for sperm penetration of the ZP in mice [105] but, rather, is responsible for dispersal of acrosomal contents during AR [106]. Furthermore, another sperm protease susceptible to p-aminobenzamidine must function as a zona-lysin in mice [107]. Bedford also claimed independently that acrosin is not involved in sperm passage of the zona pellucida but is responsible for AR and/or dispersal of the acrosomal contents during AR. He assumed that sperm penetration through the ZP might be mediated by physical sperm flagellar movement or “shear forces that oscillating thrust against the zona may create” [108]. However, flagellar movement alone appears insufficient to make a hole for the sperm passage of the ZP. In abalone, reacted sperm can penetrate the VE by the binding of sperm-derived free lysin to VERL, causing the disruption of the rod-like VERL dimer structure and the swelling of the filamentous VERL network [1,5,36]. In mammals or ascidians, however, such prominent swelling of the filamentous network has not been observed in ZP or VC by microscopic observation [1,5,36]. In the ascidian *H*. *roretzi*, the network of the VC surrounding the penetrating sperm looks, by TEM observation, chemically or enzymatically dissolved rather than physically torn by sperm movement [5]. Furthermore, most mouse sperm undergoing AR during the passage of the cumulus cell matrix before arriving at the ZP can penetrate the ZP more efficiently than the acrosome-intact sperm, as revealed by monitoring the AR with GFP-fused acrosomal protein by live imaging [95] and because mouse sperm accumulated in the perivitelline space of CD9-null eggs can penetrate the ZP and fertilize other eggs [109]. These findings indicate that zona-lysin must be localized on the surface of acrosome-reacted sperm but not released by acrosomal exocytosis. On the other hand, it was recently reported that acrosin is essential for sperm penetration of the ZP and fertilization in hamsters [110]. Live imaging data clearly demonstrated that *acrosin*-KO sperm, showing flagellar beating, can bind to the ZP but cannot penetrate the ZP [110]. Therefore, the functions of acrosin as a zona-lysin may depend on species. Although mouse acrosin does not function as a lysin, it is likely that hamster and boar acrosins function as zona-lysin. Although mouse acrosin is not essential for the sperm penetration of the ZP in mice, it is plausible that (pro)acrosin plays a role in the binding of reacted sperm to the ZP (secondary binding) [111,112,113,114]. The C-terminal propiece of proacrosin is involved in binding to ZP2 [111,112,113,114], whereas paired basic residues of acrosin heavy chain are involved in binding to the sulfated polysaccharide moiety of the ZP, as revealed by site-directed mutagenesis [115]. Taken together, sperm (pro)acrosin is involved in binding to the ZP and the dispersal of acrosomal contents, at least in mice, and the digestion of ZP proteins, at least in pigs and hamsters. Our hypothesis is that sperm (pro)acrosin-mediated attachment to and detachment from the egg coat may be necessary for the sperm passage of the egg coat in not only ascidians but also mammals.

### 6.2. Sperm Trypsin-like Proteases in H. roretzi

To investigate the sperm proteases involved in fertilization as a lysin, the effects of various protease inhibitors on fertilization were examined in *H*. *roretzi* [116]. Fertilization of intact eggs was inhibited by two chymotrypsin inhibitors (chymostatin [IC_50_ = 3 µM] and potato proteinase inhibitor 1 [IC_50_ > 100 µM]) and three trypsin inhibitors (leupeptin [IC_50_ = 20 µM], antipain [IC_50_ = 23 µM], and soybean trypsin inhibitor [IC_50_ = 35 µM]). As proteinaceous inhibitors inhibit fertilization, trypsin-like and chymotrypsin-like proteases are thought to be ectoenzymes. In addition, leupeptin at 111 µM strongly inhibited the fertilization of intact eggs (100% or 87% inhibition) but not of naked eggs (VC-removed eggs) (0% inhibition). Chymostatin at 12.3 µM potently inhibited the fertilization of intact eggs (100% inhibition) but only weakly inhibited the fertilization of naked eggs (32% or 20% inhibition). From these results, trypsin-like protease susceptible to leupeptin and chymotrypsin-like protease susceptible to chymostatin are thought to participate in sperm binding to and/or penetration of the VC. It cannot be ruled out that chymotrypsin-like protease may also be involved in another process, including gamete fusion [116].

Our laboratory examined the effects of various peptidyl-Arg (or Lys)-4-methylcoumaryl-7-amide (MCA) substrates on fertilization to explore sperm trypsin-like proteases functioning as lysins. Among the substrates tested, *tert*-butoxycarbonyl (Boc)-Val-Pro-Arg-MCA was most efficiently hydrolyzed by the enzyme in the sperm extract and most potently inhibited fertilization [117]. These results suggest that Boc-Val-Pro-Arg-MCA-hydrolyzing sperm trypsin-like protease is a candidate VC lysin. In this context, trypsin-like protease was purified from the sperm extract. By soybean trypsin inhibitor-Sepharose affinity chromatography, two trypsin-like proteases were separately purified to homogeneity [118]. Both enzymes were stabilized by a low concentration of nonionic detergent (0.005% Brig 35). A 34-kDa protease called ascidian acrosin (HrAcrosin) has a relatively broad substrate specificity toward peptidyl-Arg (or Lys)-MCA substrates, and the enzymatic properties were similar to those of mammalian acrosin. Another 28-kDa protease, designated as ascidian spermosin (HrSpermosin), is a unique protease with a very narrow substrate specificity toward Boc-Val-Pro-Arg-MCA. The inhibitory effects of various peptidyl-argininals (leupeptin analogs) on fertilization coincided well with the inhibition pattern toward HrAcrosin, except for benzyloxycarbonyl (Z)-Val-Pro-argininal, the strongest inhibitor against HrSpermosin, which exhibited stronger inhibition toward fertilization than the effects expected from the inhibitory ability toward HrAcrosin [119,120]. From these results, we concluded that not only HrAcrosin but also HrSpermosin is responsible for fertilization in *H*. *roretzi*. The anti-HrSpermosin antibody, which specifically reacted to HrSpermosin among the extracted sperm proteins, inhibited fertilization in a concentration-dependent manner. Furthermore, immunocytochemistry revealed the localization of HrSpermosin at the entire sperm head region [87,121]. HrSpermosin is partially released by the sperm reaction, but the majority of HrSpermosin was found to be associated with the sperm head region [87,121]. Localization of HrAcrosin was revealed by dansyl-Leu-argininal (a fluorescent acrosin inhibitor), showing that HrAcrosin is localized at the tip of the sperm head and mitochondrial region (see Figure 3B [unpublished data]).

Then, cDNA cloning of HrPreproacrosin [18] and HrPreprospermosin [17] was performed. Schematic diagrams of both enzyme architectures are depicted in Figure 4. HrPreproacrosin comprises a signal peptide, a short light-chain region, a heavy-chain region, and two CUB domains. In contrast, HrPreprospermosin comprises a signal peptide, a long light chain, and a heavy chain. HrProacrosin/HrOvochymase is expressed both in the testis and the FC. Rabbit acrosin has paired basic residues (Arg^50^-Arg^51^) in its N-terminal region of the heavy chain, which are involved in binding to the sulfated polysaccharide of the ZP, as described above [115]. HrProacrosin also possesses these paired basic residues (Lys^56^-His^57^), and these residues appear to be involved in its binding to the VC, as the solubilized biotinylated VC proteins can efficiently bind to the corresponding peptide (AAFLYKHVQVCG: residues 51–62) rather than the same peptide, aside from having AA instead of KH. Two peptides corresponding to the CUB1 domain (residues 348–632: TEFGVEYHTFCWYDD) and the CUB2 domain (residues 443–456: CGEFSSKHYPNYYDA) were immobilized on agarose beads and incubated with solubilized VC proteins. The CUB1-interacting VC proteins were subjected to SDS–PAGE, and five VC proteins (25-kDa, 28-kDa, 30-kDa, 85-kDa, and 95-kDa proteins) were identified. In contrast, no CUB2-interacting protein was identified. The N-terminal sequences of the CUB1-interacting proteins were determined. The longest and second longest sequences were obtained from the 30-kDa and 25-kDa proteins, respectively. Notably, the CUB1 peptide, but not the CUB2 peptide, potently inhibited fertilization. These results indicate that the HrAcrosin N-terminal residue Lys^56^-His^57^, as well as the CUB1 domain in the C-terminal propiece of HrProacrosin, are involved in binding to the VC [15]. As HrAcrosin is partially released upon sperm reaction, HrAcrosin is thought to play an extracellularly key role during fertilization, most likely as a VC lysin and a binding protein to the VC.

To identify the CUB1-interacting protein, cDNA cloning of the 30-kDa VC protein was performed. Unexpectedly, this protein was identified as a C-terminal coding region (CT) of vitellogenin, a large lipid transfer protein (LLTP) (Figure 4A). The interaction between the recombinant 30-kDa CT and the recombinant CUB1-domain protein was confirmed by a pulldown assay. The 30-kDa CT on the VC may be degraded after fertilization, as the 30-kDa band of the isolated VC disappears after the addition of sperm [16]. Consistently, the existence of the 30-kDa protein (CT) on the VC of unfertilized eggs, but not of fertilized eggs, was revealed by immunocytochemistry using an anti-30-kDa antibody [14]. Furthermore, the above degradation of CT by sperm was inhibited by leupeptin and chymostatin [16]. This suggests that the 30-kDa VC protein (CT) is degraded by sperm proteases during fertilization. On the other hand, the N-terminal sequence of the 25-kDa CUB1-interacting protein was found at the upstream region of the cDNA of the CT and was identified as a penultimate C-terminal fragment of vitellogenin (Figure 4), referred to as the von Willebrand factor D (vWF-D) region [13]. A 55-kDa protein, a precursor before processing into the 25-kDa and 30-kDa fragments, was also detected in the VC (Figure 4).

In contrast to HrProacrosin, HrSpermosin exists in two isoforms with different lengths of light chain: the type 1 isoform consists of a long light chain (L1: residues 23–129), and the type 2 isoform has a short light chain (L2: residues 97–129) (see Figure 4A) [16]. To investigate the interacting protein of HrSpermosin, GST-fusion proteins having either of these light chains (L1, L2, and L1DL2 regions) were constructed and incubated with solubilized VC proteins. Then, the interacting proteins were adsorbed to glutathione-sepharose and subjected to SDS–PAGE. The recombinant protein with L1 and L1*D*L2, but not L2, interacted with a “28-kDa” VC protein. The N-terminal sequence of this “28-kDa” protein was later found to be identical to the 25-kDa protein, a vWF-D fragment of vitellogenin. Therefore, both HrProacrosin and HrSpermosin are thought to be capable of binding to the vWF-D fragment in the VC [16]. As a long vitellogenin mRNA was abundant in the hepatopancreas, we first thought that vitellogenin must be expressed in the hepatopancreas and transferred to the oocytes during oogenesis. However, we later noticed that two short mRNAs (S1 and S2), probably alternatively spliced variants, exist in the immature oocytes and TCs during oogenesis [15]. Taken together, it is most likely that two short mRNAs of vitellogenin are expressed in the oocytes and TCs and that the vitellogenin C-terminal 25-kDa fragment (vWF-D) and 30-kDa fragment (CT) attach to the VC, to which sperm trypsin-like proteases can bind during fertilization: HrProacrosin binds to CT and vWF-D, whereas HrSpermosin binds to vWF-D (Figure 4A). We assume that sperm trypsin-like proteases are involved in not only sperm binding to the VCs but also sperm penetration of the VC because sperm cannot penetrate the VC if the binding abilities of the sperm proteases to the VC are too strong.

When protease inhibitors were added shortly after insemination, the inhibitory effects of Z-Val-Pro-Arg-H (spermosin inhibitor) and chymostatin (chymotrypsin and proteasome inhibitor) on fertilization were considerably reduced within 2 min after insemination, whereas the inhibitory ability of leupeptin (acrosin inhibitor) was almost constant even 2 min after insemination and gradually decreased until 4 min (Figure 4C) [122]. These results indicate that HrSpermosin and HrProteasome, both of which are localized on the entire sperm head region [88,121], function at the early stage during sperm penetration of the VC, whereas HrAcrosin, which is localized on the tip of the sperm head and mitochondrial region (Figure 3), plays a role until the late stage of the sperm penetration process (Figure 4D). Lambert and Koch proposed that the sperm mitochondrion functions as an anchor by attaching to the VC and that the insertion of the sperm head and tail into the perivitelline space must be driven by the mitochondrial sliding movement [23,24,96,97]. Taking into account these results, the attachment to and the proteolysis of the 30-kDa VC protein by Hr(Pro)acrosin at the surface of the mitochondrion must be necessary for sperm binding to the VC and insertion of the sperm head and tail into the perivitelline space. This must be a possible reason why the prolonged function of Hr(Pro)acrosin is necessary during sperm penetration of the VC (2–4 min).

### 6.3. Sperm Extracellular Ubiquitin–Proteasome System in H. roretzi

Because the purified sperm trypsin-like proteases showed little or no VC-degrading activity, the isolation and characterization of a sperm chymotrypsin-like protease involved in fertilization were attempted. First, we investigated the effects of four peptidyl-Tyr (or Phe)-MCA (chymotrypsin substrates) residues on fertilization [123]. Succinyl (Suc)-Leu-Leu-Val-Tyr-MCA, which was first invented in our laboratory [123], was the most potent inhibitor of fertilization. Such an inhibitory effect on the fertilization of intact eggs was considerably reduced toward the fertilization of VC-free eggs [123]. Consistently, Suc-Leu-Leu-Val-Tyr-MCA was most efficiently hydrolyzed by the enzyme in the sperm extract. These results suggest that Suc-Leu-Leu-Val-Tyr-MCA competitively inhibited fertilization, probably in the sperm passage of the VC, and that a putative sperm protease hydrolyzing Suc-Leu-Leu-Val-Tyr-MCA is involved in fertilization of *H*. *roretzi* [123].

In this context, the Suc-Leu-Leu-Val-Tyr-MCA-hydrolyzing enzyme was purified from the sperm of *H*. *roretzi* [87]. Two high-molecular-weight protease complexes were purified: one was identified as the 20S proteasome, and the other was a unique molecular species of the proteasome, designated the 930-kDa proteasome [87]. Both proteasome complexes were reacted with an anti-proteasome antibody [87], but whether the 930-kDa proteasome is identical to the 26S proteasome remains unknown. The subunit composition between the 930-kDa and the 26S proteasomes seems different, but we cannot rule out the possibility that several subunits could be dissociated by hydroxyapatite chromatography. The proteasome was partially secreted upon the sperm reaction induced by alkaline SW (pH ~ 9), where the supernatant after the treatment of the sperm with alkaline SW was called “sperm exudate”. When the sperm exudate was subjected to gel filtration, a high molecular mass (~1000-kDa) fraction showed VC-degrading activity and was reacted with an anti-proteasome antibody by dot blot analysis [123]. These results indicate that the proteasomes are partially secreted by sperm reaction and that the proteasome fraction showed a degradation activity of the VC [123]. ATP and ubiquitin (Ub), as well as the ubiquitinating enzyme complex, were also released upon sperm reaction [86]. Thus, all of the factors necessary to accomplish the novel extracellular Ub-proteasome system (UPS) are secreted upon sperm reaction in ascidian sperm.

Generally, intracellular abnormal proteins or short-lived proteins are tagged with Ub via the isopeptide bond between the C-terminal Gly residue of Ub and the Lys residue of the substrates; this is catalyzed by the sequential actions of the Ub-activating enzyme E1, the Ub-conjugating enzyme E2, and the Ub ligase E3. Thus, ubiquitinated proteins are degraded by the 26S proteasome, which comprises the 20S proteasome and the 19S regulatory particle (19S RP) [124,125,126,127]. The 20S proteasome is a 700-kDa multicatalytic protease complex with a barrel-shaped structure comprising four stacked heptameric rings of α and β subunits (α_7_, β_7_, β_7_, α_7_ subunits), where the β1, β2, and β5 subunits have caspase-like, trypsin-like, and chymotrypsin-like activity, respectively. In contrast, 19S RP comprises 19 regulatory subunits, i.e., 6 ATPase subunits (Rpt1–6) and 13 non-ATPase subunits (Rpn1–15: missing Rpn4 and Rpn14) [127]. ATP is required for ubiquitination, the assembly of the 26S proteasome, and the unfolding of the substrate polypeptide. The ubiquitinated substrates are recognized by Rpn10 and Rpn13, followed by unfolding by ATPases (Rpt1–6) and degradation by the β1, β2, and β5 subunits. Although the intracellular roles of 26S proteasomes are well known [124,125,126,127], the roles of extracellular proteasomes or membrane-bound proteasomes are only poorly understood.

Our laboratory first discovered that the sperm proteasome plays a key role in the sperm penetration of the VC, particularly in the degradation of the VC, in *H*. *roretzi* [45,87,88]. We noticed that the main component of the VC, HrVC70, is degraded by the sperm extract and the purified 26S proteasome in the presence of Ub, ATP, and the ubiquitinating enzymes (E1, E2, E3) [45,88]. As described previously, HrVC70 is a self/nonself-recognizable sperm receptor on the VC (Figure 3). There are two Lys residues in HrVC70, of which Lys^234^ was identified as a ubiquitinated residue by site-directed mutagenesis (see Figure 3) [45]. HrVC70 is ubiquitinated during fertilization, as revealed by Western blot analysis using a membrane reprobed with anti-HrVC70 antibody and an FK2 monoclonal antibody that specifically reacts to mono-Ub and multi-Ub chains but not free Ub. These results coincided with the results of immunostaining with an FK2 antibody, showing that the VC is ubiquitinated during fertilization [45]. Importantly, the FK2 monoclonal antibody can inhibit the fertilization of *H*. *roretzi*, which indicates the requirement of extracellular ubiquitination for fertilization [45]. The 20S proteasome may also be involved in sperm binding to the VC based on the results comparing the inhibitory abilities of leupeptin analogs toward proteasomal activities and sperm binding to the VC [128]. However, such sperm binding to the VC of glycerol-treated eggs might be mediated by ES complex formation.

This extracellular ubiquitinating enzyme complex (exUbEC) was purified from sperm exudate by chromatography with DEAE-cellulose and Ub-agarose and by glycerol-gradient centrifugation [86]. Ubiquitination assays were performed using HrVC70 as a substrate and [^125^I] Ub, MgATP, and MG115 (a proteasome inhibitor). The purified exUbEC can ubiquitinate HrVC70 but not heat-denatured lysosomes, indicating a highly specific reaction. The molecular mass of exUbEC was estimated to be 700 kDa by glycerol gradient centrifugation, indicating that E1, E2, and E3 form a complex similar in size to that of the 20S proteasome. Notably, exUbEC can play an extracellular role in SW because the optimum pH is approximately the pH of SW (pH ~ 8); furthermore, the activity requires a high concentration of Ca^2+^ (~10 mM), a concentration near that of SW but not the intracellular concentration, and is very active under high concentrations of Na^+^ (~0.4 M) near the SW concentration. We also discovered that apyrase, an ATP-diphosphohydrolase, can inhibit both the exUbEC-mediated ubiquitination of HrVC70 and the fertilization by depleting extracellular ATP. Although it remains unknown whether apyrase mainly inhibits the extracellular ubiquitination of HrVC70, the degradation of HrVC70, or other processes during fertilization, the fact that extracellular ATP is necessary for ascidian fertilization was our novel and amazing discovery. Similar results were obtained in the other animals, as described below.

The reasons why we believe that the extracellular UPS participates in the sperm passage of the VC in ascidians are summarized as follows: (1) the proteasome inhibitors (MG115 and MG132) inhibited fertilization; (2) the anti-proteasome and anti-HrVC70 antibodies inhibited fertilization; (3) the sperm proteasome located at the sperm head was activated and partially secreted by sperm reaction (sperm activation), as judged using Suc-Leu-Leu-Val-Tyr-MCA; (4) the proteasome-enriched fraction in the gel filtration of sperm exudate showed activity in degrading the VC; (5) the proteasome was partially partitioned in the hydrophobic (membrane) fraction using Triton X-116 and can be purified from the sperm membrane fraction; (6) the purified sperm 26S proteasome can degrade ubiquitinated HrVC70; (7) the FK2 monoclonal antibody, which is specific to the mono- and the multi-Ub chain, can inhibit fertilization; (8) the HrVC70-specific ubiquitinating enzyme complex (exUbEC) is secreted upon sperm reaction; (9) Lys^234^ of HrVC70 is ubiquitinated by exUbEC; (10) immunocytochemistry using the FK2 antibody showed that VC is ubiquitinated during fertilization; (11) proteasome, exUbEC, ATP, and Ub are secreted by the sperm reaction; and (12) the depletion of extracellular ATP by apyrase potently inhibited fertilization. Although it is impossible to knock out proteasomal subunit genes because of essential genes, testis-specific knockout or the knockdown of these genes may clarify the roles of the sperm proteasome in fertilization in the future.

To search for a potential sorting signal of the proteasome to the surface of the sperm head, the 20S proteasome was purified from ascidian sperm, eggs, and muscles, and the subunit compositions were compared by 2D-PAGE [129]. Then, we noticed that the spot mobility of the α6/PSMA1 subunit of the sperm 20S proteasome is different from those of the egg and muscle 20S proteasomes. Using LC/MS/MS analysis, we revealed that the C-terminal 16 residues of the α6/PSMA1 subunit of the sperm 20S proteasome are processed [129]. As there are acidic amino acid clusters in the C-terminal 16 residues, removal of the C-terminal 16 residues may affect the function of the proteasome. Alternatively, a newly appeared C-terminal sequence may afford a new signal for sorting to some organelle or membrane fraction. In fact, it is intriguing to note that the C-terminal tripeptide PTS-1 and nanopeptide PTS-2 are known to be signals for transport to peroxisomes [130,131,132]. In this system, even a large protein complex can pass through the peroxisome membrane. This putative unknown sorting signal may be involved in translocation of the proteasome to the membrane or extracellular milieu.

### 6.4. Sperm Extracellular Ubiquitin–Proteasome System in Sea Urchins and Vertebrates

In sea urchins, the effects of various microbial protease inhibitors on fertilization were examined by Hoshi and his colleagues [133]. As chymostatin inhibited sea urchin fertilization, chymotrypsin-like protease was assumed to be a VE-lysin [133]. Then, a chymotrypsin-like protease was purified and characterized by Aketa’s laboratory [134,135]. The enzyme has the ability to dissolve the VE. After treatment of the eggs with the purified chymotrypsin-like protease, the fertilized eggs became 2- and 4-cell embryos without a fertilization membrane, a proteinaceous coat derived from the VE. However, the purified enzyme was potently inhibited by 0.1 mM chymostatin, but the sea urchin fertilization was not inhibited by chymostatin at 0.1 mM [133]. This is a contradictory point that remains to be solved. On the other hand, Aketa’s group reported that the proteasome is involved in AR in the sea urchin [136]. Our laboratory attempted to identify the sperm proteases involved in sea urchin fertilization by examining the effects of various proteasome inhibitors, including proteasome inhibitors, on fertilization. Among the protease inhibitors at 0.1 mM, MG132 most potently inhibited fertilization (96% inhibition), and MG115 (32% inhibition) and lactacystin (22% inhibition) showed moderate or weak inhibition, whereas chymostatin, leupeptin, soybean trypsin inhibitor, and E-64-d showed no appreciable inhibition (less than 3% inhibition) [93,137]. Among the proteasome substrates, Z-Leu-Leu-Glu-MCA most potently inhibited fertilization, and Boc-Leu-Arg-Arg-MCA weakly inhibited fertilization at 0.1 mM, whereas Suc-Leu-Leu-Val-Tyr-MCA showed no appreciable inhibition at 0.1 mM [93]. Egg-jelly-induced AR, but not Ca^2+^ ionophore-induced AR, was inhibited by MG132, suggesting that the proteasome is involved in the process before the Ca^2+^ influx induced by the egg jelly. Our results coincided well with previous findings by Matsumura and Aketa [136]. On the other hand, MG132 showed no inhibition toward the binding of acrosome-reacted sperm to the VE, suggesting that the proteasome is not involved in sperm binding to the VE. Furthermore, the inhibition pattern of various protease inhibitors toward the three catalytic centers of the proteasome was compared with the inhibition pattern toward fertilization. The results indicate that the caspase-like activity of the proteasome must be involved in fertilization, particularly in sperm penetration of the VE. The 20S proteasome subunits were detected in the acrosomal contents by Western blotting, indicating that the proteasome is included in an acrosome. Sea urchin egg VE appears to be ubiquitinated before fertilization, as revealed by immunocytochemistry and Western blot analysis using an FK2 monoclonal antibody. Apyrase also inhibited fertilization, suggesting the necessity of extracellular ATP in fertilization, similar to ascidians. Taken together, sea urchin sperm proteasome is involved in fertilization, most likely in the sperm penetration of the VE and in the AR.

In mammals, Sutovsky and his colleagues elucidated that the UPS is involved in sperm passage of the ZP by digesting the ZP components but not in sperm binding to the ZP [138,139,140,141]. The boar sperm proteasome is located at the acrosome, as shown by immunocytochemistry and by using a transgenic pig expressing a GFP-tagged PSMA1/a6 subunit-containing proteasome. The fluorescence due to the GFP-proteasome was monitored under fluorescence microscopy [138,141]. According to several lines of evidence, the sperm proteasome is localized in the acrosome and is involved in digestion of the ZP [140]. Similarly, it has been reported in quail (bird) that the sperm proteasome is involved in the passage of the egg coat [136]. Therefore, the sperm extracellular proteasome system plays a key role in ascidians, sea urchins, birds, and mammals. This appears to be a common mechanism in fertilization shared by deuterostomes.

The proteasome is reportedly involved in sperm capacitation and AR in mammals [140,142,143,144]. There are several reports about the relationship between reduced proteasome activity and abnormalities or low motility in sperm [145]. Furthermore, some male infertility patients have anti-sperm antibodies against sperm plasma membrane proteins, among which two proteins were identified as proteasome subunits [146]. On the other hand, in addition to seven α subunits in the house-keeping 20S proteasome, there is a testis-specific subunit α4s/PSMA8 in mammals [147]. Notably, this subunit is specifically expressed in the testis and is involved in histone degradation during spermiogenesis [148,149,150]. PSMA8-null spermatocytes show delayed M-phase entry and arrest at this stage, resulting in male infertility [150]. Similar, but not identical, testis-specific proteasomal α subunits have also been reported in fruit flies [151]. Furthermore, the proteasome activators PA28γ/PSME3 [152] and PA200/PSME4 [152,153] have also been reported to participate in spermatogenesis by gene-targeting experiments. In particular, the double KO of PSME3 and PSME4 causes complete male infertility in mice [152]. Because S4α-expressing proteasome and proteasome activators play key roles in spermiogenesis and spermatogenesis, gene-targeting experiments to determine the functions of sperm proteasomes in fertilization would not be easy to perform.

### 6.5. Ascidian Sperm Proteases: Discovery of Astacin-like Metalloprotease

To examine whether the sperm proteasome and the chymotrypsin-like protease, which are proposed to be a VC-lysin [154,155,156], are localized at the sperm cell surface, our laboratory performed a proteomic analysis of the sperm surface protein-enriched fraction, which was obtained by the treatment of intact and reacted sperm with cell-impermeable sulfo-NHS-LC biotin, followed by isolation with NeutrAvidin agarose in *C*. *intestinalis* [40]. Unexpectedly, the chymotrypsin-like protease or proteasome subunits were not detected under the conditions tested. Instead, several peaks of astacin-like metalloproteases and thrombospondin-type-1 repeats, which we named “Tast (tunicate astacin-like metalloprotease with thrombospondin-type-1 repeat)”, were identified [157]. There are five *Tast* genes in *C**. intestinalis***:**
*Tast1* on chromosome 9 and four contiguous *Tast* genes (*Tast2a*, *Tast2b*, *Tast2c*, and *Tast2d*) on chromosome 1. The five Tasts are type II transmembrane proteins with a single transmembrane domain but not a signal peptide at the N-terminal region (we failed to identify a transmembrane domain in our previous report [157]). These five genes are transcribed in the testis, and the proteins are expressed in spermatozoa. The involvement of metalloproteases in fertilization was examined by the metalloprotease inhibitor GM6001. GM6001 strongly inhibited the fertilization of intact eggs but not VC-free eggs, and GM6001 did not inhibit sperm binding to the VC of glycerinated eggs. Furthermore, when isolated VC was incubated with intact sperm, several VC proteins, including CiVC57, were degraded and were inhibited by GM6001. Therefore, the Tasts are thought to be promising candidates for VC lysin, although we cannot rule out the possibility that the 24-kDa chymotrypsin-like protease and the proteasome are also involved in sperm penetration of the VC in *C*. *intestinalis*. To obtain direct evidence for the participation of Tasts in the sperm penetration of the VC, a KO of the *Tast* genes was attempted [157] by genome editing. Although adult ascidians were not obtained due to abnormal metamorphosis, we noticed that the hatching of tadpole larva was delayed. Interestingly, several egg coat-targeting proteases, such as mouse egg ovastacin [158,159], and two medaka hatching enzymes, choliolysin H (HCE) [160] and choliolysin L (LCE) [161], are astacin-like metalloproteases [162]. In addition, hatching enzymes from sea urchin [163,164] and *C*. *intestinalis* [165] are also metalloproteases. Mouse ovastacin is known to hydrolyze the N-terminal region of ZP2, resulting in a polyspermy block via the loss of sperm receptor activity [159]. These results also support our conclusion that metalloproteases are also involved in the digestion of VC in ascidians. Interestingly, the purified medaka hatching enzymes HCE and LCE can hydrolyze Suc-Leu-Leu-Val-Tyr-MCA most efficiently among the peptidyl-MCA substrates tested [160,161]. Usually, metalloproteases cleave the N-terminal-side peptide bond of hydrophobic residues such as Leu. However, HCE and LCE can hydrolyze the C-terminal side of the Tyr residue. Therefore, whether the inhibitory effect of Suc-Leu-Leu-Val-Tyr-MCA on ascidian fertilization may be due to the effect on sperm Tast activities in addition to the proteasomes should be reexamined.

## 7. Gamete Fusion in Ascidians

In ascidians, the molecular mechanism of gamete fusion is not known. It was proposed that a metalloprotease is involved in gamete fusion because Ca^2+^ chelators or metalloprotease substrates inhibited the fusion between sperm and eggs [166]. However, to the best of our knowledge, GM6001, an astacin-like metalloprotease inhibitor, potently inhibited the fertilization of intact eggs but not the fertilization of VC-free eggs in *Ciona intestinalis* [157] and *Halocynthia roretzi* (unpublished data). Therefore, although membrane fusion events may require Ca^2+^ or other metal ions, this does not necessarily mean that metalloprotease is involved in gamete fusion. Rather, our results using GM6001 did not support their assumption.

In mice, sperm inner acrosomal membrane-bound IZUMO1 [167], an immunoglobulin-like membrane protein, sperm acrosome associated 6 (SPACA6) [168], transmembrane protein 95 (Tmem95) [169,170], and dendrocyte expressed seven transmembrane protein domain-containing (DCST) 1 and 2 [171] are essential for sperm–egg fusion. The IZUMO1 receptor JUNO [172], an egg plasma membrane folate receptor 4, is necessary for sperm-egg adhesion. The IZUMO1-JUNO complex is the first pair of gamete recognition factors to be determined at an atomic resolution [173,174]. In addition, egg CD9 is also indispensable for sperm–egg fusion, as CD9 gene-KO eggs cannot fuse with sperm [175]. CD9 plays a crucial role in the microvilli biogenesis and molecular organization, such as JUNO localization, of the oocyte surface [176,177]. Interestingly, CD9 is reported to be transferred to the sperm membrane with the aid of egg-derived esoxomes [178]. CD9-transferred sperm can fuse with CD9-deficient eggs [178]. To the best of our knowledge, homologous genes to these essential genes for gamete fusion have not been identified in ascidians. In flowering plants, GCS1/HAP2 is known to be a male gamete-specific protein responsible for gamete fusion [179,180]. This gene has been identified in not only flowering plants but also cnidarians and phylogenetically lower animals and plasmodium [180,181]. However, homologous genes have not been identified in ascidians. Notably, *GCS1***/***HAP2* expression is always restricted to the male gamete, even in isogamy. Interestingly, GCS1/HAP2 is structurally similar to the class II viral membrane fusion proteins [182]. There may be common molecules and common mechanisms shared by animals and plants. Furthermore, the molecular mechanisms of gamete fusion might be much more similar to those of viral fusion or somatic cell fusion than we previously thought.

## 8. Polyspermy Block in Ascidians

Two polyspermy block mechanisms have been proposed in sea urchin eggs [1]: one is a fast block to polyspermy, which is triggered by the rapid depolarization of membrane potential, and the other is a late block to polyspermy, in which the cortical granule proteases released upon gamete fusion degrade the sperm receptor on the VE and the connecting bridge between the VE and the egg plasma membrane. Recently, it was claimed that the fast polyspermy block mediated by Na^+^-influx-induced depolarization is a misinterpretation because low-Na^+^ SW impaired the elevation of the fertilization membrane, but these eggs are still monospermic, and apparent abnormal cleavage is due not to polyspermy but rather to alterations in the cortical actin filament dynamics [183]. It is also claimed by Dale that the idea of a fast block to polyspermy is unfounded [184]. However, it is plausible that cortical granule proteases play a key role in the degradation of sperm receptors and the cleavage of the connection between VE and the egg plasma membrane in sea urchins [1].

In mouse, ZP2 is reported to be a species-specific sperm receptor [1]. After gamete fusion, astacin-like metalloprotease, named ovastacin, is released from cortical granules and hydrolyzes the N-terminal region of ZP2, leading to the inactivation of the sperm receptor. Notably, mouse spermatozoa cannot bind to the ZP of normal 2-cell embryos, but they can bind to the ZP of ovastacin-null 2-cell embryos or ZP2-mutant 2-cell embryos, whose ZP2 cannot be cleaved by ovastacin. These results indicate that the digestion of ZP2 by cortical granule ovastacin is responsible for the establishment of the polyspermy block in mice [159].

In ascidians, it is proposed that glycosidase(s) released from egg cells or FCs upon fertilization are responsible for the blockage of polyspermy. As described in Section 2, FC glycosidase is thought to hydrolyze or bind to the carbohydrate moiety of the sperm receptor on the VC [21]. Generally, the optimum pH values of glycosidases are approximately 4–6, and the enzyme shows little or no hydrolyzing activity in SW conditions (pH ~ 8.2). Our laboratory showed that GlcNAc-specific lectins strongly inhibited the fertilization of *H*. *roretzi* among the various lectins tested [185]. These results suggest that the GlcNAc moiety of VC glycoproteins may function as a sperm receptor on the VC [185]. Among various fluorogenic substrates for glycosidases, GlcNAcase and GalNAcase (*N*-acetylgalactosaminidase) are most highly released from eggs upon activation of eggs with calcium ionophore A23187. This GlcNAcase (or preferably HexNAcase: N-actylhexosaminidase) was purified to homogeneity from *H*. *roretzi* eggs. The purified enzyme gave a single band with a molecular mass of 56 kDa after deglycosylation on SDS–PAGE. The molecular mass was estimated to be 520 kDa by gel filtration, suggesting the oligomeric nature of this enzyme. This enzyme can bind to the VC. From these results, it is thought that GlcNAcase is released from eggs (or FCs) upon fertilization and binds to the sperm receptor on the VC, participating in the blockage of polyspermy. Sperm α-L-fucosidase is reportedly involved in sperm binding to the VC as the specific antibody inhibits fertilization [55]. Sperm α-L-fucosidase may be involved in sperm binding to the VC, and the GlcNAcase released from eggs upon fertilization may be responsible for the polyspermy blockade by binding to the sperm receptor. The most promising candidate for the sperm receptor in *H*. *roretzi* is HrVC70. As described in Section 4, HrVC70 contains monofucosylated Thr^362^ in EGF7 and Ser^143^ in EGF-3. Furthermore, Thr^254^ in EGF5 and Ser^470^ in EGF9 are partially glycosylated with trisaccharides such as Galβ1→4GlcNAcβ1→3Fucα1→O-Ser/Thr. Therefore, sperm α-L-fucosidase may be capable of binding to fucosylated Ser^143^, Thr ^254^, Thr^362^, and Ser ^470^, but such binding may face interference by the binding of egg-derived GlcNAcase to Thr ^254^ and Ser ^470^ on the VC. Sperm binding to the VC must be reduced by this mechanism.

## 9. Conclusions and Perspectives

We have been using *H*. *roretzi* and *C*. *intestinalis* for fertilization research. The candidate molecules involved in the gamete interactions and lysins are summarized in Table 1 and Figure 2. Hoshi et al. first examined the effects of various protease inhibitors on the fertilization of *H*. *roretzi*. Then, Sawada et al. purified two trypsin-like proteases from *H*. *roretzi* sperm and showed their binding abilities to the C-terminal region of vitellogenin on the VC. Vitellogenin is believed to be a lipid transfer protein, but two C-terminal fragments produced by proteolysis function as binding proteins for sperm trypsin-like protease: they were amazing discoveries for us. In addition, the sperm proteasome was found to be released by sperm reaction or sperm activation upon sperm binding to the VC. Ub and Ub-conjugating enzyme complexes with a molecular mass of 700 kDa (exUbEC) were also found to be released by sperm reactions. These findings are also unexpected as there have been no reports showing that the proteasome or Ub-conjugating enzyme complex functions extracellularly. However, it is currently believed that extracellular UPSs function similarly to lysins during the fertilization of deuterostomes, including mammals, birds, ascidians, and sea urchins. In addition, evidence of the participation of the extracellular UPS in the nervous system, cell migration, lung injury and repair, and many other pathological and physiological systems is accumulating [186,187,188,189,190]. In addition, self/nonself recognition systems appear to be very similar to the self-incompatibility system in angiosperms. These findings were also unexpected. Ascidian fertilization research has taught us a generality rather than a particularity in fertilization mechanisms and has showed us the occurrence of common mechanisms in fertilization shared by animals and plants. However, our knowledge is still very limited. The 3D structural analyses of gamete recognition proteins as well as genetic analyses will answer many questions that remain to be elucidated in the future.

## Figures and Tables

**Figure 1 cells-11-02096-f001:**
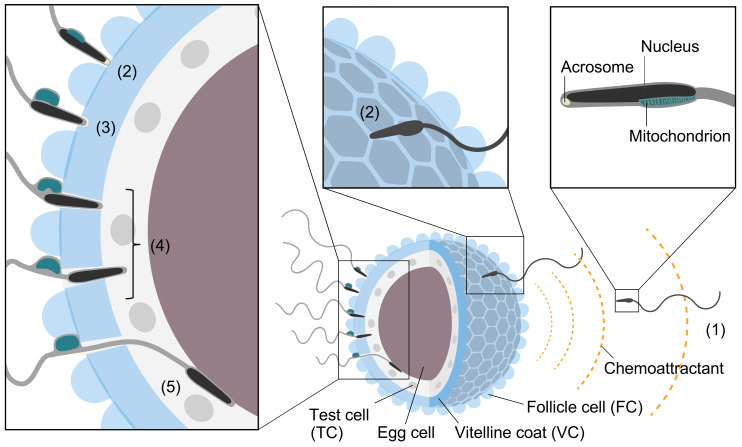
Fertilization processes of the ascidian *Halocynthia roretzi*. Generally, fertilization processes consist of 5 steps: (1) sperm activation and chemoattraction; (2) sperm binding to the egg coat (species and self/nonself recognition); (3) sperm reaction (sperm activation and AR); (4) sperm penetration of the egg coat (vitelline coat (VC)) with the aid of lysin; and (5) gamete fusion. In ascidians, upon sperm binding to the VC, the sperm undergoes a sperm reaction, which is characterized by vigorous movement on the VC and mitochondrial swelling and sliding through the tail. In *H*. *roretzi*, a self/nonself-gamete-recognizable sperm receptor on the surface of the VC, called VC70, is exposed between FCs. It is proposed that the sperm head and tail are inserted into the perivitelline space by the driving force of the sliding of the mitochondrion, whose plasma membrane is attached to the VC. The sperm (pro)acrosin located at the surface of the mitochondrion may be involved in binding to the VC. In this process, the ubiquitin–proteasome system is partially released or exposed on the sperm surface ubiquitinates and degrades VC70, which is necessary for sperm penetration of the VC. Sperm astacin-like metalloprotease is also involved in this process.

**Figure 2 cells-11-02096-f002:**
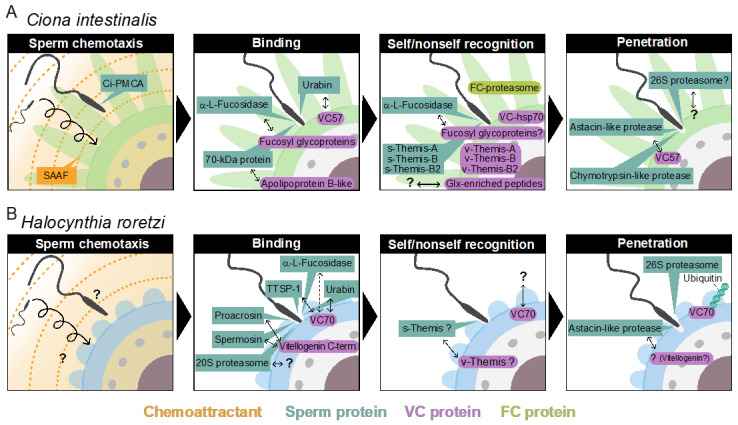
Fertilization processes of *C*. *intestinalis* and *H*. *roretzi*. (**A**) Fertilization processes in *Ciona intestinalis*. Sperm are activated and chemoattracted to the eggs. Then, the sperm binding to the VC is mediated by the sperm CiUrabin and α-L-fucosidase. The binding partners on the VC are CiVC57 and fucosyl-glycoproteins. Upon sperm binding, self/nonself recognition is carried out by sperm Ci-s-Themis-A, B, and B2 and the same haplotypic recognition partners on the VC and Ci-v-Themis-A, B, and B2. In the penetration process, sperm chymotrypsin-like protease, astacin-like protease, and the 26S proteasome appear to function as a lysin or lysin system. (**B**) Fertilization processes in *Halocynthia roretzi*. Sperm binding to VC is mediated by sperm HrUrabin, HrTTSP-1, α-L-fucosidase, HrProacrosin, and HrSpermosin. The binding partner of HrUrabin, HrTTSP-1, and α-L-fucosidase on the VC is thought to be HrVC70. The binding partners on the VC of HrProacrosin and HrSpermosin (type 1) are C-terminal fragments (25-kDa and 30-kDa fragments) of vitellogenin. After the sperm recognizes the VC as nonself, the sperm ubiquitin–proteasome system must be activated, resulting in ubiquitination of HrVC70 by a 700-kDa ubiquitinating enzyme complex released upon sperm reaction, which is subsequently degraded by the sperm 26S proteasome. This allows sperm penetration of the VC. Meanwhile, Hr(Pro)acrosin is also involved in sperm binding and penetration. Sperm astacin-like metalloprotease also participates in the sperm penetration process of the VC. The target of astacin-like protease may be vitellogenin on the VC (unpublished data).

**Figure 3 cells-11-02096-f003:**
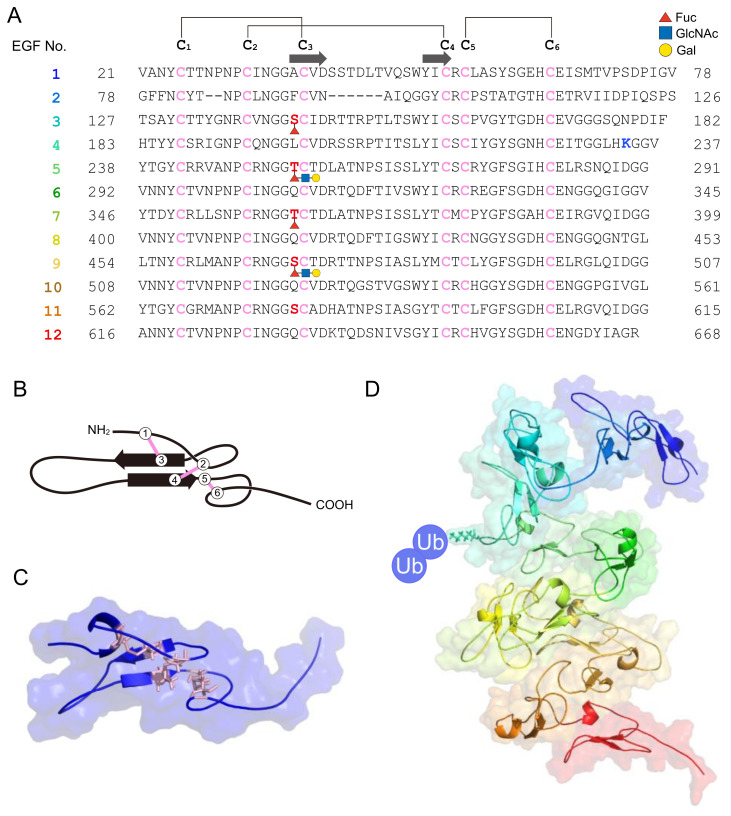
Structure of the 70-kDa-vitelline-coat protein HrVC70, a self/nonself-recognizable sperm receptor in the ascidian *Halocynthia roretzi*. (**A**) Amino acid sequence of HrVC70 and consensus sequences for *O*-fucosylation. Conserved Cys residues are indicated by magenta. Potential fucosylation sites and ubiquitinated sites are indicated by red and blue, respectively. Carbohydrate modifications were investigated by LC/MS/MS, and possible glycosylations were indicated. (**B**,**C**) Positions of disulfide bridges and anti-parallel β-sheets of the EGF1 domain are illustrated. (**D**) The 3D structure of HrVC70 was predicted by AiphaFold2. Lys^234^, which is ubiquitinated during fertilization, is indicated by the blue ball-and-stick model attaching di-Ub.

**Figure 4 cells-11-02096-f004:**
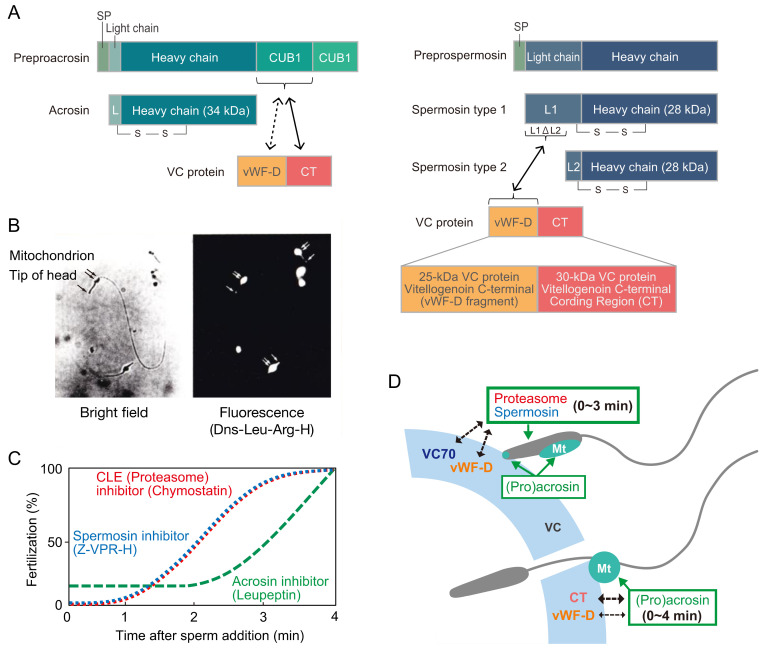
(**A**) Domains of *H*. *roretzi* sperm trypsin-like proteases and their binding proteins. (**B**) Localization of Hr acrosin. Acrosin is revealed by the fluorescent-labeled acrosin inhibitor dansyl-leucyl-argininal (Dns-Leu-Arg-H). Strong fluorescence of the dansyl group was observed at the tip of the sperm head (single arrow) and mitochondrial region (double arrows) attached to the side of the sperm head. (**C**) Functioning time of sperm proteases. (**D**) Working hypothesis for the roles and timing of sperm proteases.

**Table 1 cells-11-02096-t001:** Candidate molecules involved in gamete interactions and lysins in *C*. *intestinalis* and *H*. *roretzi*.

Fertilization Process	*Ciona intestinalis* (Type A)		*Halocynthia roretzi*	
	Sperm Factor	Egg Factor	Sperm Factor	Egg Factor
**Sperm chemotaxis**	PMCA	Steroidal saponin (3,4,7,26-tetrahydroxy- cholestane-3,26-disulfate)	ND	ND
**Sperm binding to VC**	Urabin	VC57	Urabin	VC70
	α-L-Fucosidase	140-kDa, 125-kDa, and 78-kDa fucosyl glyco- protein on VC surface	TTSP-1	VC70
			α-L-Fucosidase	VC70?
	70-kDa protein	Apolipoprotein B-like	Proacrosin	30-kDa CT (C-terminal of VG), 25-kDa vWF-D (C-terminal of VG)
			Spermosin	25-kDa vWF-D (C-terminal of VG)
			20S Proteasome	ND
**Self/nonself recognition** **(Acquisition of self-sterility during oocyte maturation)**		Autologous peptides produced by FC-proteasome are attached to FC-hsp70		VC120 is processed to VC70 by FC-TLE (FC-Ovochymase?)
**Self/nonself recognition (During fertilization)**	s-Themis-A	v-Themis-A	ND	VC70
	s-Themis-B	v-Themis-B	s-Themis?	v-Themis?
	s-Themis-B2	v-Themis-B2		
	TLE?	v-Themis-like		
	α-L-Fucosidase	140-kDa, 125-kDa, and 78-kDa fucosyl glycoprotein on VC surface		
	ND	Glx-enriched protein		
**Sperm VC lysins**	26S proteasome	ND	Extracellular 700-kDa Ubiquitinatingenzyme complex	VC70
			Extracellular 26S proteasome	VC70
	Astacin-like metalloprotease	VC57	Astacin-like metalloprotease	ND (VG ?)
	Chymotrypsin-like protease	ND		

Abbreviations: CT, C-terminal coding region; FC, follicle cell; PMCA, plasma membrane calcium/calmodurin-dependent calcium ATPase; ND, not determined; TLE, trypsin-like enzyme; VC, vitelline coat; VG, vitellogenin; vWF, von Willebrand factor-D.

**Table 2 cells-11-02096-t002:** Male and female determinants in self-incompatibility of plants and animals.

Family	Female Determinant	Male Determinant	Mode of Self/Nonself Recognition
Angiosperms (Plants)			
Brassicaceae	SRK	SP11/SCR	Self recognition
Papaveraceae	PrsS	PrpS	Self recognition
Solanaceae/Rosaceae	S-RNase	SLF/SFB	Nonself recognition
Urochodate (Animals)			
Cionidae (Phlebobranchia, e.g., *Ciona intestinalis*)	v-Themis-A, B, B2	s-Themis-A, B, B2	Self recognition
Pyuridae (Stolidobranchia, e.g., *Halocynthia roretzi*)	VC70	(TTSP1, Urabin)?	Nonself recognition?

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
