# Peer review of "Mechanisms of Sperm–Egg Interactions: What Ascidian Fertilization Research Has Taught Us"

_cells, 2022, doi:10.3390/cells11132096_

Round 1

Reviewer 1 Report

The introduction is too short and reductive. Sexual reproduction starts with gametogenesis and virtually ends at the first zygote division and fertilization is one aspect of the process. Authors should expand the scientific background of the studies reported (lines 28-28). Each step in fact should be better detailed especially for the not expert readership of Cells. As an example, acrosome reaction is poorly described   whereas is a fundamental step in the fertilization process. This is of further importance in ascidians whereas a not clear and ascertained acrosome reaction occurs and most of the studies on this topic date back to 40 years ago (line 123).  

Line 31 instead of “are not well known” should write “are not yet fully clarified”

Line 32 here is missing the first step which is the motility induction. Authors should mention the intricate process of reciprocal gamete activation which starts when the extracellular egg coat induces motility in the quiescent sperm cell. This step is then followed by chemotaxis, binding etc.. (see Tosti and Menezo, 2016).

Paragraph 8. Polyspermy is a subject very controversial and still a matter of debate. By introducing this subject it is worth mentioning the important contribution of B. Dale to this issue.

In conclusion:

This review is very interesting, well written and the authors are experts on the matter and have ongoing investigations on the same topic. This review provides accurate and detailed information on the molecular mechanisms that underlie main ascidian fertilization steps reporting also useful and appropriate comparisons with other species such as the sea urchin and some mammalian models. The figures are well designed and the references updated. I have particularly appreciated the detailed explanation of paragraph 5.1 on Ciona self- sterility and the following ones on other species. This topic has been studied in the 80’ providing some important information on the biological mechanisms at the basis of immunology.  

I believe that this review is worthy of publication in Cells after minor revisions mainly concerning some aspects of the scientific background of the subject.

Author Response

Dear Editors of Cells,

Response to Reviewer 1 Comments.

(Point 1)

The introduction is too short and reductive. Sexual reproduction starts with gametogenesis and virtually ends at the first zygote division and fertilization is one aspect of the process. Authors should expand the scientific background of the studies reported (lines 28-28). Each step in fact should be better detailed especially for the not expert readership of Cells. As an example, acrosome reaction is poorly described whereas is a fundamental step in the fertilization process. This is of further importance in ascidians whereas a not clear and ascertained acrosome reaction occurs and most of the studies on this topic date back to 40 years ago (line 123). 

(Response 1)

Thank you for your valuable comments. Since the detailed mechanisms in each step are described in later sections, detailed explanations in each fertilization step were not included in “Introduction” section. However, as “Introduction” part appears to be very short, Sections 1 and 2 were combined. As suggested, we did not mention about acrosome reaction (AR), since ascidian sperm acrosome is too small to detect by TEM and also since the timing of AR is still debated.

However, as acrosome reaction is an important step, we inserted a new section after a section of “Self/nonself-recognition”, entitled “Acrosome reaction (AR) and sperm reaction”. In this section, we briefly summarized the basic mechanisms in sea urchins, starfishes and mouse. In addition, the controversial points in ascidian AR and a unique morphological change called “sperm reaction” were mentioned in this section. For detail, please see the attached files (indicated by yellow highlight).

(Point 2)

Line 31: Instead of “are not well known” should write “are not yet fully clarified”

(Response 2)

Line 31: “… are not well known.” was modified to “… are not yet fully clarified.”

(Point 3)

Line 32 here is missing the first step which is the motility induction. Authors should mention the intricate process of reciprocal gamete activation which starts when the extracellular egg coat induces motility in the quiescent sperm cell. This step is then followed by chemotaxis, binding etc.. (see Tosti and Menezo, 2016).

(Response 3)

Line 32: “… sperm chemoattraction …”  ---> “… sperm activation and chemoattraction …”.

In addition, a reference by “Tosti and Menezo (2016)” was cited in the revised version.

Line 14: “… spermatozoa are first attracted to …” was modified to  “… spermatozoa are first activated and attracted to …” 

Line 130: Section title was changed to “Ascidian sperm activation and chemotaxis”

(Point 4)

Paragraph 8. Polyspermy is a subject very controversial and still a matter of debate. By introducing this subject it is worth mentioning the important contribution of B. Dale to this issue.

(Response 4)

Line 1032: “….actin filament dynamics [xxx]. However, …” has been changed to

“….actin filament dynamics [183]. It is also claimed by Dale that the idea of a fast block to polyspermy is unfounded [184]. However, …”

After the paragraph describing the sea urchin polyspermy block, the following paragraph, describing the mechanism of mouse polyspermy block, was added, since Dean’s discovery by clear experiments is worth-mentioning.

“In mouse, ZP2 is reported to be a species-specific sperm receptor [1]. After gamete fusion, astacin-like metalloprotease, named ovastacin, is released from cortical granules and hydrolyzes the N-terminal region of ZP2, leading to the inactivation of sperm receptor. Notably, mouse spermatozoa cannot bind to the ZP of normal 2-cell embryos, but they can bind to the ZP of ovastacin-null 2-cell embryos or ZP2-mutant 2-cell embryos, whose ZP2 cannot be cleaved by ovastacin. These results indicate that digestion of ZP2 by cortical granule ovastacin is responsible for the establishment of polyspermy block in mice [159].

Reviewer 2 Report

Fertilization is key for generating a new terrestrial organism and pivotal for genetic diversity. Several steps must occur for gamete fusion and fertilization to occur. Initially, egg-derived molecules attract animal sperm to the egg. As sperm pass through the egg’s extracellular matrix or when interacting with the egg coat, the cell undergoes the acrosome reaction, an exocytotic process. Self/nonself recognition in hermaphrodites, i.e. ascidians, occurs when sperm bind to the egg coat. In ascidians egg coat penetration by activated or acrosome-reacted sperm involves the extracellular ubiquitin–proteasome system, astacin-like metalloproteases and trypsin-like proteases. The authors summarize gamete recognition and the role of egg coat lysins in ascidian fertilization, in animals and in plants.

In general, I found the review very interesting and well written. However, I found some sections to be a bit too detailed for general readers interested in Reproductive Biology. Though I understand it to be difficult, and a matter to be decided by the Editor and authors, it would be good to summarize more some sections that I indicate in the pdf which I will attach (see suggested corrections and some questions in blue and in the margins).

1)     Page 2, line 69. In a review like this it might be better to deal at the beginning with confusions between ovum and egg. In some texts it is indicated that an egg is a fertilized ovum, so?

2)     In Fig. 1 where or what is the change associated with the acrosome reaction (AR) in the illustration?

3)     On the top of page 3, the link between the transient Ca2+ elevation due to the decrease in PMSA and SOC activation is not clear at all.

4)     The section on Gamete binding in Halocynthia roretzi seems too detailes. In addition, it seems what should be in italics in the subtitle is Halocynthia roretzi and not Gamete binding in. I might be wrong as I see it happens in other subtitles.

5)     Section 6.3. Sperm extracellular ubiquitin–proteasome system in H. roretzi, could be summarizes as the authors already did starting in line 854, and numbered 1-12.

Author Response

Thank you very much for your valuable comments and suggestions. We modified according to the comments and suggestions by reviewer 2, as follows. On the other hand, we would not like to modify the original wording, since English grammar was proofread by 3 editors in American Journal Expert (AJE) before submission.

(Point 1)

Page 2, line 69. In a review like this it might be better to deal at the beginning with confusions between ovum and egg. In some texts it is indicated that an egg is a fertilized ovum, so?

(Response 1)

Generally, in mammals, ovum or oocyte is used in many papers. But in marine invertebrates, “egg” is a popular term. So, in line 70, we changed “an ovum (egg cell)” to “an egg cell”, because so-called “eggs” mean not only egg cells but also test cells, vitelline coat, and follicle cells in ascidians. In order to distinguish them, we used “egg cell” in Figure 1 and line 70.

Line 70: “an ovum (egg cell) is …”   ”an egg cell is ..”

(Point 2)

In Fig. 1 where or what is the change associated with the acrosome reaction (AR) in the illustration?

(Response 2)

Thanks for your suggestions. Acrosome indicated by “white small vesicle” was newly depicted in step 2 but not in step 3 in Figure 1. In step (3), acrosome is missing due to accomplishment of acrosome rection. 

(Point 3)

On the top of page 3, the link between the transient Ca2+ elevation due to the decrease in PMSA and SOC activation is not clear at all.

(Response 3)

SOC is always involved in Ca2+ influx, whereas PMSA is involved in Ca2+ efflux. Since sperm is circulating, sperm can detect the descending gradient of SAAF. At the lowest concentration of SAAF, PMSA activity must be minimum because of lowest concentration of SAAF. As a result, Ca2+ influx overcome Ca2+ efflux, resulting in the increase in intracellular Ca2+ concentration. In the text, we slightly modified some expression for better understanding.

In order to clarify our explanation, some words (indicated by yellow) were added in the revised manuscript. In addition, circulating sperm was illustrated in Fig. 2A(1st step) and 2B (1st step).

(Point 4)

The section on Gamete binding in Halocynthia roretzi seems too detailes. In addition, it seems what should be in italics in the subtitle is Halocynthia roretzi and not Gamete binding in. I might be wrong as I see it happens in other subtitles.

(Response 4)

Some detailed description such as the procedure of Far western blotting (line 287-291) was deleted. 6th paragraph of section 6.3 (line 836-852) was also deleted to shorten the text.

Concerning the subtitle (subheading), the style was unified according to the style of “template” of this journal. Subheading should be in italic. In this case, italic letters such as species name should be indicated by regular letter.

(Point 5)

Section 6.3. Sperm extracellular ubiquitin–proteasome system in H. roretzi, could be summarizes as the authors already did starting in line 854, and numbered 1-12.

(Response 5)

In this review article, we described the historical flow and our unexpected discovery. So, in addition to the summary, we would like to keep mentioning the entire story of our discovery. 

Round 2

Reviewer 1 Report

the authors have correctly answered to my comments and the ms has been improved. It is now acceptable in its present form

Reviewer 2 Report

Overall the authors have addressed the major points made by the reviewers, therefore the paper is ready to be accepted.